



**Reduced growth with increased quotas of particulate organic and inorganic**
**carbon in the coccolithophore *Emiliania huxleyi* under future ocean climate**
**change conditions**
**Yong Zhang,[1,4] Sinéad Collins,[2] Kunshan Gao[1,3,*]**
[1]State Key Laboratory of Marine Environmental Science and College of Ocean and
Earth Sciences, Xiamen University, Xiamen, China
[2]Institute of Evolutionary Biology, School of Biological Sciences, University of
Edinburgh, Edinburgh EH9 3FL, United Kingdom
[3]Co-Innovation Center of Jiangsu Marine Bio-industry Technology, Jiangsu Ocean
University, Lianyungang, China
[4]College of Environmental Science and Engineering, and Fujian Key Laboratory of
Pollution Control and Resource Recycling, Fujian Normal University, Fuzhou, China
Running head: Response of *E. huxleyi* to multiple drivers
[*]Correspondence: Kunshan Gao (ksgao@xmu.edu.cn)
Keywords: $CO_2$; coccolithophore; functional trait plasticity; light; multiple drivers;
nutrients; ocean acidification; warming.



**Abstract**
Effects of ocean acidification and warming on marine primary producers can be
modulated by other environmental factors, such as levels of nutrients and light. Here,
we investigated the interactive effects of five oceanic environmental drivers ($CO_2$,
temperature, light, dissolved inorganic nitrogen and phosphate) on growth rate,
particulate organic (POC) and inorganic (PIC) carbon quotas of the cosmopolitan
coccolithophore *Emiliania huxleyi*. Population growth rate increased with increasing
temperature (16 to 20 $^o$C) and light intensities (60 to 240 µmol photons m$^{-2}$ s$^{-1}$), but
decreased with elevated $p$$CO_2$ concentrations (370 to 960 µatm) and reduced
availability of nitrate (24.3 to 7.8 µmol L$^{-1}$) and phosphate (1.5 to 0.5 µmol L$^{-1}$). POC
quotas were predominantly enhanced by combined effects of increased $p$$CO_2$ and
decreased availability of phosphate. PIC quotas increased with decreased availability
of nitrate and phosphate. Our results show that concurrent changes in nutrient
concentrations and $p$$CO_2$ levels predominantly affected growth, photosynthetic carbon
fixation and calcification of *E. huxleyi*, and imply that plastic responses to progressive
ocean acidification, warming and decreasing availability of nitrate and phosphate
reduce population growth rate while increasing cellular quotas of particulate organic
and inorganic carbon of *E. huxleyi*, ultimately affecting coccolithophore-related
ecological and biogeochemical processes.



## 1 Introduction

Ocean acidification (OA), due to continuous oceanic absorption of anthropogenic $CO_2$, is occurring alongside ocean warming. This in turn, leads to shoaling in the upper mixed layer (UML) and a consequent reduction in the upward transport of nutrients into the UML. These ocean changes expose phytoplankton cells within the UML to multiple simultaneous stressors or drivers, and organismal responses to these drivers can affect both trophic and biogeochemical roles of phytoplankton (see reviews by Boyd et al., 2015; Gao et al., 2019 and literatures therein). While most studies on the effects of ocean global climate changes on marine primary producers have focused on organismal responses to one, two or three environmental drivers, there is an increasing awareness of the need to measure the combined effects of multiple drivers (see reviews by Riebesell and Gattuso, 2015; Boyd et al., 2018; Gao et al., 2019; Kwiatkowski et al., 2019). For this purpose, several manipulative experimental approaches have been recommended (Boyd et al., 2018). One approach using many unique combinations of different numbers of drivers showed that both short and long-term growth responses were, on average, explained by the dominant single driver in a multi-driver environment, but this result relies on having many (>5) drivers with known or measured large-effect single drivers (Brennan and Collins, 2015; Brennan et al., 2017). For experiments with multiple drivers where interactions are likely to preclude making predictions from single drivers, where average responses are not the most informative ones, or where logistics preclude using a very large number of multi-driver environments, Boyd et al. (2010) suggested an 'environmental cluster' method where key drivers (such as temperature, light intensity, nutrient concentration, $CO_2$ and Fe) are covaried within experiments, allowing the investigation of physiological responses of phytoplankton to concurrent changes of the clustered



drivers. This approach examines responses to projected overall environmental shifts
rather than pulling apart the biological or statistical interactions between responses to
individual drivers. To our knowledge, studies to date have employed such a driver
clustering approach to investigate responses of diatoms *Fragilariopsis cylindrus*,
*Thalassiosira pseudonana*, *Skeletonema costatum*, and the prymnesiophyte
*Phaeocystis antarctica* to combinations of drivers projected for 2100 (Xu et al., 2014a;
Xu et al., 2014b; Boyd et al., 2016).
An environmental cluster approach is especially useful when drivers are known to
interact in terms of the organismal responses they elicit, as is the case for OA, light
levels, and key nutrients acting on population growth rate and carbon fixation (Boyd
et al., 2016). For example, in the cosmopolitan coccolithophore *Emiliania huxleyi*,
interactive effects of OA and light showed that OA increased population growth rate
and photosynthetic carbon fixation under low light, whereas it slightly lowered
population growth rate and photosynthetic carbon fixation under high light
(Zondervan et al., 2002; Kottmeier et al., 2016). In addition, photosynthetic carbon
fixation was further enhanced by longer light exposure at high $p$CO$_2$ levels
(Zondervan et al., 2002). On the other hand, OA can exacerbate the negative impact
of solar UV radiation on photosynthetic carbon fixation and calcification in *E. huxleyi*
under nutrient-replete conditions (Gao et al., 2009), but can increase calcification
(coccolith volume) and particulate organic carbon (POC) quota under phosphate-
limited conditions (Leonardos and Geider, 2005; Müller et al., 2017), demonstrating
that the effects of OA on calcification is likely nutrient-dependent. On the other hand,
ocean warming, which occurs alongside OA, is known to increase coccolith length,
POC, particulate organic nitrogen (PON) and inorganic carbon (PIC) production rates
of several *E. huxleyi* strains (Rosas-Navarro et al., 2016; Feng et al., 2017). Warming



has also been shown to increase the optimal $p\text{CO}_2$ levels for growth, POC and PIC
production rates (Sett et al., 2014). In one case warming was found to compensate for
the negative impact of OA on growth rate under low light intensity (Feng et al., 2008).
Nevertheless, decreased photosynthetic carbon fixation and calcification at reduced
carbonate saturation state (lowered $\text{Ca}^{2+}$ concentrations) were exacerbated by
warming treatment (Xu et al., 2011). Overall, there is strong evidence that
understanding the plastic responses of this key calcifier to ocean changes requires
investigating responses to the overall expected shift in the environment, in addition to
the detailed studies to date on individual drivers, due to the sheer number of
interactions between individual drivers on traits that affect the trophic and
biogeochemical roles of *E.huxleyi.*
Despite known interactions among two- and three-way combinations of OA,
temperature, light, phosphate levels and nitrogen levels, there have been few
empirical studies investigating effects of the larger cluster projected for future surface
ocean changes. The data to date show that interactions among drivers can affect both
the direction and magnitude of trait changes in biogeochemically important taxa. In
addition, based on single or two-driver studies, changes in temperature, $p\text{CO}_2$, light,
dissolved inorganic nitrogen (DIN) and phosphate (DIP) in combination are predicted
to affect primary productions (Barton et al., 2016; Monteiro et al., 2016; Boyd et al.,
2018; Gao et al., 2019; Kwiatkowski et al., 2019). Understanding the trait-based
responses of cocolithophores to future ocean changes is important for projections of
changes in the biogeochemical roles of phytoplankton, such as biological carbon
pump efficiency (Rost and Riebesell, 2004).
In order to understand the combined effects of $p\text{CO}_2$, temperature, light, dissolved
inorganic nitrogen (DIN) and phosphate (DIP) on functional traits, we incubated





*Emiliania huxleyi* (Lohmann) under different combinations of environmental
conditions that represented subsets of, and eventually the complete set of
environments for, this environmental driver cluster. We recently examined the
interactive effects of light intensity and $CO_2$ level on growth rate, POC and PIC
quotas of *E. huxleyi* under nutrients replete, low DIN, or low DIP concentrations
(Zhang et al., 2019). Light, $CO_2$, DIN and DIP levels usually change simultaneously
with temperature, and temperature modulated responses of *E. huxleyi* to other
environmental drivers (Gafar and Schulz, 2018; Tong et al., 2019). In addition,
warming or cooling can direcly influence the activity of enzymes, thus directly
modulating metabolic rates (Sett et al., 2014). Because of the overwhelming evidence
that temperature can act as a general modulator of organismal responses, we use the
present study to examine how the addition of temperature as a key driver in the
environmental change cluster can modulate the combined effects of $CO_2$, light and
nutrients. We found that future ocean scenario treatments with OA, warming,
increased light and reduced availability of nutrients led to lower growth rate and
larger POC and PIC quotas of *E. huxleyi*.

**2 Materials and Methods**
**2.1 Experimental setup**
*Emiliania huxleyi* strain PML B92/11 was originally isolated from coastal waters off
Bergen, Norway, and obtained from the Plymouth algal culture collection, UK. The
average levels of $p$$CO_2$, temperature, light, dissolved inorganic nitrate (DIN) and
phosphate (DIP) were set up according to recorded data in Norwegian coastal waters
during 2000 to 2007 and projected for 2100 in high-latitudes (Larsen et al., 2004;
Locarnini et al., 2006; Omar et al., 2010; Boyd et al., 2015) (Table S1). *E. huxleyi* was



cultured with a 12 h/12 h light/dark cycle in thermo-controlled incubators in Aquil
medium, which was prepared according to Sunda et al. (2005) with the addition of
2200 μmol $L^{-1}$ bicarbonate to achieve the total alkalinity (TA) of 2200 μmol $L^{-1}$. The
experiment was conducted in five steps (Fig. 1). Considering ocean acidification and
warming as the key drivers for ocean climate changes, we first established 4 "baseline"
treatments where the $p$CO$_2$ and temperature drivers were combined in a fully factorial
way: low $p$CO$_2$ + low temperature (LCLT), high $p$CO$_2$ + low temperature (HCLT),
low $p$CO$_2$ + high temperature (LCHT), and high $p$CO$_2$ + high temperature (HCHT).
Since reduced availability of nutrients and increased light exposures are triggered by
warming-enhanced stratification, we then added additional single or pairs of drivers to
each of these "baseline" treatments (Fig. S1). In step 1, low light (LL) was added; in
step 2, high light (HL) was added. HL was then maintained for the rest of the
experiment. In step 3, low nitrogen was added and high phosphate levels were
maintained (LNHP). In step 4, low phosphate was added and high nitrogen levels
were restored (HNLP). In step 5, both nitrogen and phosphate were low (LNLP),
respectively (Figs. 1 and S1). In all cases, the cells were acclimated to each unique
stressor cluster for at least 14–16 generations before physiological and biochemical
parameters were measured. Although this stepwise design introduces a historical
effect, physiological traits are generally reported after 10 to 20 generations
acclimation to OA treatment (Perrin et al., 2016; Tong et al., 2016; Li et al., 2017), so
the historical effects here are similar to those that would be introduced with standard
methods in other physiology studies (Tong et al., 2016; Zhang et al., 2019). Since
individually reduced availability of nitrate or phosphate decreased growth, did not
change POC quota, and enhanced PIC quota under optimal light intensity (HL in this
study) in the same *E. huxleyi* strain (Zhang et al., 2019), we hypothesized that



combination of DIN and DIP limitation would result in similar trend under the $p$CO$_2$
and/or temperature combined treatments. Therefore, we added stepwise nitrate and/or
phosphate drivers (Fig. 1).
For step 1, NO$_3^-$ and PO$_4^{3-}$ were modified to 24 µmol L$^{-1}$ and 1.5 µmol L$^{-1}$,
respectively, which is the HNHP treatment in the synthetic seawater (Sunda et al.,
2005) (Fig. S1). The seawater was dispensed into 4 glass bottles, and 2 bottles of
seawater were placed at 16 $^{\circ}$C (LT) in an incubator (HP400G-XZ, Ruihua, Wuhan),
and aerated for 24 h with filtered (PVDF 0.22 µm pore size, Haining) air containing
400 µatm (LC) or 1000 µatm $p$CO$_2$ (HC). Another 2 bottles of seawater were
maintained at 20 $^{\circ}$C (HT) in the other chamber and also aerated with LC or HC air as
described above. The dry air/CO$_2$ mixture was humidified with deionized water prior
to the aeration to minimize evaporation. The LCLT, HCLT, LCHT and HCHT
seawaters (Figs. 1a and S1) were then filtered (0.22 µm pore size, Polycap 75 AS,
Whatman) and carefully pumped into autoclaved 250 mL polycarbonate bottles
(Nalgene, 4 replicate flasks for each of LCLT, HCLT, LCHT and HCHT, a total of 16
flasks at the beginning of the experiment) with no headspace to minimize gas
exchange. The flasks were inoculated at a cell density of about 150 cells mL$^{-1}$. The
volume of the inoculum was calculated (see below) and the same volume of seawater
was taken out from the bottles before inoculation. The samples were initially cultured
at 60 µmol photons m$^{-2}$ s$^{-1}$ (LL) of photosynthetically active radiation (PAR)
(measured using a PAR Detector, PMA 2132 from Solar Light Company) under
LCLT, HCLT, LCHT and HCHT conditions for 8 generations (6 days) (d), and then
the samples were diluted to their initial concentrations and grown for another 8
generations (6 d) (Fig. 1a). Samples in culture bottles were mixed twice a day at 9:00
a.m. and 5:00 p.m. At the end of the incubation, sub-samples were taken for



measurements of cell concentration, POC and TPC quotas, TA, pH and nutrient
concentrations.
In step 2, samples grown under the previous conditions were transferred at the end
of the cultures from 60 (LL) to 240 μmol photons $m^{-2}$ $s^{-1}$ (HL) of PAR with initial
cell concentrations of 150 cells $mL^{-1}$, and acclimated to the HL for 8 generations (5 d
in 16 $^{o}$C environment, 4 d in 20 $^{o}$C environment) (Fig. 1b). The cultures were then
diluted to achieve initial cell concentration and incubated at the HL for another 8
generations (the fifth day in 16 $^{o}$C environment and the fourth day in 20 $^{o}$C
environment) before sub-samples were taken for measurements.
In step 3, step 4 and step 5, $NO_3^-$ and $PO_4^{3-}$ concentrations were set to be 8 μmol $L^-$
$^1$ and 1.5 μmol $L^{-1}$ for the LNHP treatment, and 24 μmol $L^{-1}$ and 0.5 μmol $L^{-1}$ for the
HNLP treatment, and 8 μmol $L^{-1}$ and 0.5 μmol $L^{-1}$ for the LNLP treatment,
respectively (Fig. 1c,d,e). The LCLT, HCLT, LCHT and HCHT were step 1
conditions, now we are into step 3, 4 and 5. Under 240 μmol photons $m^{-2}$ $s^{-1}$ (HL) of
PAR, cell samples with an initial concentration of 150 cells $mL^{-1}$ were transferred
from HNHP condition (step 2) to LNHP conditions (step 3) and acclimated to LNHP
conditions for 8 generations (5 d in 16 $^{o}$C environment, 4 d in 20 $^{o}$C environment)
(Fig. 1c). The cultures were then diluted back to initial cell concentrations and
incubated in the LNHP conditions (step 3) for a further 8 generations. On the last day
of the incubation (the fifth day in 16 $^{o}$C environment and the fourth day in 20 $^{o}$C
environment), sub-samples were taken for measurements of the parameters.
After that, cell samples were transferred stepwise from HNHP conditions (step 2,
Fig. 1b) to HNLP conditions (step 4, Fig. 1d), then from HNLP conditions to LNLP
conditions (step 5, Fig. 1e). Cell samples were acclimated for 8 generations at HNLP
and LNLP conditions, respectively, and followed by another 8 generation incubations





for 4 d at HT and 5 d at LT. On the fourth day (for populations in high temperature
environments) or the fifth day (for populations in low temperature environments),
sub-samples were taken for measurements (Fig. 1d,e). At low nutrient concentrations,
maximal cell concentrations were limited by nutrients (Rouco et al., 2013; Rokitta et
al., 2016). To check whether cells sampled were in exponential growth at each
nutrient level, we examined cell concentrations every day at LCHT, or LCLT and
high light conditions (Fig. S2). We found that cell concentrations were in the
exponential growth phase during the $1^{st}$ and $5^{th}$ days at HT, and during the $1^{st}$ and $7^{th}$
days at LT. In this study, we taken samples in the $4^{th}$ day at HT and in the $5^{th}$ day at
LT, and thus cells sampled were in the exponential growth phase of *E. huxleyi*.

In the previous work (Zhang et al., 2019), we transferred *E. huxleyi* cells stepwise

from 80 μmol photons $m^{-2}$ $s^{-1}$ to 120 μmol photons $m^{-2}$ $s^{-1}$, then to 200 μmol photons
$m^{-2}$ $s^{-1}$, to 320 μmol photons $m^{-2}$ $s^{-1}$ and to 480 μmol photons $m^{-2}$ $s^{-1}$ at both LC and
HC levels under HNHP, LNHP or HNLP conditions, respectively. In this study, we
transferred the same strain from LL to HL under HNHP condition, and then from
HNHP to LNHP or HNLP, and from HNLP to LNLP under HL conditions under 4
"baseline" $CO_2$ and temperature treatments, in an effort to elucidate interactive and
combined effects of temperature, $CO_2$, DIN and DIP (Table S2), in contrast the
previous work carried out under constant temperature (Zhang et al., 2019).

**2.2 Nutrient concentrations and carbonate chemistry measurements**
In the first and last days of the incubations, 20 mL samples for determination of
inorganic nitrogen and phosphate concentrations were taken at the same time using a
filtered syringe (0.22 μm pore size, Haining) and measured by using a scanning
spectrophotometer (Du 800, Beckman Coulter) according to Hansen and Koroleff



(1999). The nitrate was reduced to nitrite by zinc cadmium reduction and then total
nitrite concentration was measured. In parallel, 25 mL samples were taken for
determination of total alkalinity (TA) after being filtered (0.22 μm pore size, Syringe
Filter) under moderate pressure using a pump (GM-0.5A, JINTENG) and stored in the
dark at 4 $^{o}$C for less than 7 d. TA was measured at 20 $^{o}$C by potentiometric titration
(AS-ALK1+, Apollo SciTech) according to Dickson et al. (2003). Samples for pH$_{T}$
(total scale) determinations were syringe-filtered (0.22 μm pore size), and the bottles
were filled from bottom to top with overflow and closed immediately without
headspace. The pH$_{T}$ was immediately measured at 20 $^{o}$C by using a pH meter
(Benchtop pH, Orion 8102BN) which was calibrated with buffers (Tris•HCl, Hanna)
at pH 4.01, 7.00 and 10.00. Carbonate chemistry parameters were calculated from TA,
pH$_{T}$, phosphate (at 1.5 μmol L$^{-1}$ or 0.5 μmol L$^{-1}$), temperature (at 16 $^{o}$C or 20 $^{o}$C), and
salinity using the $CO_2$ system calculation in MS Excel software (Pierrot et al., 2006).
K$_{1}$ and K$_{2}$, the first and second carbonic acid constants, were taken from Roy et al.

(1993).


**2.3 Cell concentration measurements**
In the last day of the incubation, ~25 mL samples (8 samples) were taken at the same
time (about 1:00 p.m.). Cell concentration and cell diameter (D) were measured using
a Z2 Coulter Particle Count and Size Analyzer (Beckman Coulter). The diameter of
detected particles was set to be 3 to 7 μm in the instrument, which excludes detached
coccoliths (Müller et al., 2012). Cell concentration was also measured by microscopy
(ZEISS), and variation in measured cell concentration between two methods was ±
7.9% (Zhang et al., 2019). Average growth rate ($\mu$) was calculated for each replicate
according to the equation: $\mu = (\ln N_1 - \ln N_0) / d$, where $N_0$ was 150 cells mL$^{-1}$ and $N_1$





was the cell concentration in the last day of the incubation, $d$ was the growth period in
days. *E. huxleyi* cells were spherical and its cell volume with coccoliths was
calculated according to the equation: $V = 3.14 \times (4/3) \times (D/2)^3$.

**2.4 Total particulate (TPC) and particulate organic (POC) carbon measurements**
100 mL samples for determination of TPC and POC quotas were filtered onto GF/F
filters (pre-combusted at 450 $^o$C for 6 h) at the same time in each treatment. TPC and
POC samples were stored in the dark at –20 $^o$C. For POC measurements, samples
were fumed with HCl for 12 h to remove inorganic carbon, and samples for TPC
measurements were not treated with HCl. All samples were dried at 60 $^o$C for 12 h,
and analyzed using a Thermo Scientific FLASH 2000 CHNS/O elemental analyzer
(Thermo Fisher, Waltham, MA). Particulate inorganic carbon (PIC) quota was
calculated as the difference between TPC quota and POC quota. POC and PIC
production rates were calculated by multiplying cellular contents with $\mu$ (d$^{-1}$),
respectively. Variations in measured carbon content between the four replicates were
calculated to be 1–24% in this study.

**2.5 Data analysis**
Firstly, we examined the interactions of temperature, $p$CO$_2$ and light under nutrient-
replete (HNHP) conditions. Here, the effects of temperature, $p$CO$_2$, light intensity and
their interaction on growth rate, POC and PIC quotas were tested using a three-way
analysis of variance (ANOVA). Secondly, we examined the effects of nutrient
limitation in the different $p$CO$_2$ and temperature environments under the high light
intensity (HL). Here, the effects of temperature, $p$CO$_2$, dissolved inorganic nitrogen
(DIN), dissolved inorganic phosphate (DIP) and their interaction on growth rate, POC



and PIC quotas were tested using a four-way ANOVA. Finally, a one-way ANOVA
was used to test the differences in growth rate, POC and PIC quotas between present
(defined as low levels of $p$CO$_2$, temperature and light along with high levels of DIN
and DIP (LC LT LL HN HP)) and future ocean (defined as higher levels of $p$CO$_2$,
temperature, and light along with low levels of DIN and DIP (HC HT HL LN LP))
scenarios. A Tukey post hoc test was performed to identify the differences between
two temperatures, two $p$CO$_2$ levels, two DIN or two DIP treatments. Normality of
residuals was conducted with a Shapiro-Wilk's test, and a Levene test was conducted
graphically to test for homogeneity of variances. A generalized least squares (GLS)
model was used to stabilize heterogeneity if variances were non-homogeneous. All
statistical calculations were performed using $R$ (R version 3.5.0).
In order to quantify the individual effect of nitrate concentration or phosphate
concentration on the physiological and biochemical parameters, we calculated the
change ratio ($R$) of physiological rates according to the equation: $R = \mid M_{\text{LNHP or HNLP}}$
$- M_{\text{HNHP}} \mid / M_{\text{HNHP}}$, where $M_{\text{LNHP or HNLP or HNHP}}$ respresents measured trait values in
LNHP or HNLP or HNHP conditions, and the '$\mid$' denotes the absolute value
(Schaum et al., 2013). We then calculated the expected growth rate, POC quota and
PIC quota in LNLP conditions based on the measured trait values in HNHP
conditions and the change ratios in LNHP and HNLP conditions according to a linear
model: $E_{\text{LNLP}} = (1 - R_{\text{LNHP}} - R_{\text{HNLP}}) \times M_{\text{HNHP}}$ for growth rate and POC quota; $E_{\text{LNLP}} =$
$(1 + R_{\text{LNHP}} + R_{\text{HNLP}}) \times M_{\text{HNHP}}$ for PIC quota (Brennan and Collins, 2015). We tested the
significant differences between the expected trait values ($E_{\text{LNLP}}$) and the measured
trait values ($M_{\text{LNLP}}$) in LNLP conditions by a one-way ANOVA (Fig. S3). We also
calculated the extent of synergy between LNHP and HNLP on growth rate, POC





quota and PIC quota according to equation: $S = \mid E_{\mathrm{LNLP}} - M_{\mathrm{HNHP}} \mid / M_{\mathrm{HNHP}}$. Please
see the discussion section for more information.

**3 Results**
**3.1 Carbonate chemistry parameters and nutrient concentrations**
During the incubations, $pH_T$ values increased due to organismal activity by, on
average, $0.03 \pm 0.01$ in LCLT, by $0.01 \pm 0.01$ in HCLT, by $0.02 \pm 0.01$ in LCHT and
by $0.02 \pm 0.01$ in HCHT conditions (Fig. 1f–j; Table 1). Correspondingly, seawater
$p\mathrm{CO}_2$ concentrations decreased by $8.8\% \pm 1.1\%$ in LCLT, by $6.1\% \pm 4.4\%$ in HCLT,
by $6.6\% \pm 1.7\%$ in LCHT, and by $5.4\% \pm 3.6\%$ in HCHT conditions, respectively
(Fig. 1k–o; Table 1).
During the incubations, dissolved inorganic nitrogen (DIN) concentrations
decreased by $28.7\% \pm 6.7\%$ in HNHP and LL (Fig. 1p), by $26.8\% \pm 5.9\%$ in HNHP
and HL (Fig. 1q), by $71.1\% \pm 3.3\%$ in LNHP (Fig. 1r), by $32.9\% \pm 5.6\%$ in HNLP
(Fig. 1s), and by $69.8\% \pm 3.2\%$ in LNLP conditions (Fig. 1t; Table 2). Dissolved
inorganic phosphate (DIP) concentrations decreased by $62.2\% \pm 16.5\%$ in HNHP and
LL (Fig. 1u), by $71.3\% \pm 6.7\%$ in HNHP and HL (Fig. 1v), by $61.0\% \pm 5.2\%$ in
LNHP (Fig. 1w), by $83.8\% \pm 5.4\%$ in HNLP (Fig. 1x), and by $86.3\% \pm 1.4\%$ in LNLP
conditions (Fig. 1y; Table 2).
Overall, while organismal activity affected nutrient levels during growth cycles as
expected, the high and low nutrient treatments remained different at all times (Table
2). Organismal activity had minimal effects on carbonate chemistry (see Fig. 1).

**3.2 Population growth rate**





Growth rate was significantly lower under the future scenario (HCHT HL LNLP: high
levels of $p\text{CO}_2$, temperature and light as well as low levels of nutrients) than under the
present scenario (LCLT LL HNHP: low levels of $p\text{CO}_2$, temperature and light
alongside high levels of nutrients) (one-way ANOVA, F = 52.6, $p < 0.01$) (Figs. 2a
and 3a,d; Table 2). The effect of increasing $p\text{CO}_2$ on growth rate is negative at low
light or low nutrients levels, which can be seen by comparing population growth in all
of the HC regimes with their paired LC regimes (Figs. 3a,b,e and S4). The extent of
reduction in population growth rate depends on which other stressors are present.
Compared to present atmospheric $p\text{CO}_2$ levels (LC, Fig. 3a), growth rates under ocean
acidification (HC, Fig. 3b) decreased by an average of 17.4% ± 1.3% in HNHP and
LL, and by an average of 4.4% ± 1.1% in HNHP and HL conditions (three-way
ANOVA, both $p < 0.01$; Tukey post hoc test, both $p < 0.01$) (Fig. 3e; Tables 2 and 3),
by 7.6% ± 2.6% in LNHP, by 21.4% ± 0.2% in HNLP, and by 32.1% ± 0.5% in
LNLP conditions under the HL, respectively (four-way ANOVA, all $p < 0.01$; Tukey
post hoc test, all $p < 0.01$) (Fig. 3a,b,e; Tables 2 and 4).

Across all HT/LT (high/low temperature) regime pairs, population growth rate is

faster in the HT regimes, indicating that increasing temperature from 16 to 20 $^{\circ}$C
increases population growth rate in *E. huxleyi* (Figs. 3a,c,f and S4). Compared to the
low temperature (LT, Fig. 3a), growth rates at the high temperature (HT, Fig. 3c)
increased by 7.7% ± 0.7% in HNHP and LL, and by 34.0% ± 0.4% in HNHP and HL
conditions (three-way ANOVA, both $p < 0.01$; Tukey post hoc test, both $p < 0.01$)
(Fig. 3a,c,f; Tables 2 and 3), by 42.4% ± 0.4% in LNHP, by 33.5% ± 0.5% in HNLP,
and by 40.4% ± 3.1% in LNLP conditions under HL (four-way ANOVA, all $p < 0.01$;
Tukey post hoc test, all $p < 0.01$) (Fig. 3a,c,f; Tables 2 and 4). Compared to low $p\text{CO}_2$
and low temperature (LCLT, Fig. 3a), growth rates in high $p\text{CO}_2$ and high





temperature environments (HCHT, Fig. 3d) increased by 3.9% ±0.9% in HNHP and
LL, and by 31.1% ±0.1% in HNHP and HL conditions (three-way ANOVA, both $p <$
0.01; Tukey post hoc test, both $p < 0.01$) (Fig. 3a,d,g; Tables 2 and 3), by 38.6% ±0.1%
in LNHP and by 17.1% ±1.7% in HNLP, whereas growth rate decreased by 12.1% ±
2.2% in LNLP conditions under HL, respectively (four-way ANOVA, all $p < 0.01$;
Tukey post hoc test, all $p < 0.01$) (Fig. 3a,d,g; Tables 2 and 4). These results show
that high $p$CO$_2$, low nitrate and low phosphate concentrations collectively reduced the
population growth rate in *E. huxleyi*, though elevated temperature could counteract
this response.

The effects of reduced availability of nutrients on growth are nutrient-specific (Fig.

3). Compared to HNHP and HL, growth rates in LNHP decreased by 3.0–12.1% (all $p$
$< 0.05$ at LCLT, HCLT, LCHT and HCHT conditions) (Fig. 3h; Tables 2 and 4). In
contrast, HNLP did not significantly affect growth in LC conditions ($p > 0.1$ in LCLT
and LCHT conditions) (Fig. 3a,c,i), but did lower population growth rate by 11.3–
19.2% in HC conditions (both $p < 0.01$ at HCLT and HCHT conditions) (Fig. 3b,d,i).
Unsurprisingly, when both nitrate and phosphate levels were reduced, growth rates
always decreased by larger extent compared to environments where they were
reduced individually (Fig. 3h,i,j). Compared to growth rates in HNHP and HL, growth
rates in LNLP were 4.8–10.2% lower in LC environments, and 34.7–40.3% lower in
HC environments (Tukey post hoc test, all $p < 0.01$ at LCLT, HCLT, LCHT and
HCHT conditions) (Fig. 3a–d,j; Tables 2 and 4). In summary, nitrate and phosphate
limitation exacerbated the impacts of OA and warming on population growth rate.

**3.3 POC quota**





Cellular POC quotas were two-fold larger under the future scenario (HCHT HL LNLP)
than under the current scenario (LCLT LL HNHP) (one-way ANOVA, F = 96.1, $p <$
0.01, Figs. 2b and 4a,d). The effect of increasing $p\mathrm{CO_2}$ on POC quota is positive,
regardless of other drivers present, which can be seen by comparing POC quotas in all
of the HC regimes with their paired LC regimes (Figs. 4a,b,e and S4), though the
extent of increase in POC quota depends on which other stressors are present.
Compared to current atmospheric $p\mathrm{CO_2}$ level (LC, Fig. 4a), POC quotas under ocean
acidification (Fig. 4b) increased by 40.3% ± 10.1% in HNHP and LL (Tukey post hoc
test, $p < 0.01$), by 13.8% ± 10.1% in HNHP and HL ($p = 0.47$), by 33.2% ± 11.1% at
LNHP, by 109.4% ± 14.0% in HNLP and by 87.3% ± 10.8% in LNLP conditions
under HL, respectively (four-way ANOVA, all $p < 0.01$; Tukey post hoc test, all $p <$
0.01) (Fig. 4a,b,e; Tables 2 and 4).
The effect of elevated temperature on POC quota can be seen by comparing POC
quota in all of the HT regimes with their paired LT regimes (Figs. 4a,c,f and S4).
Across all HT/LT regime pairs, POC quotas did not show significant differences
between the HT and LT regimes under HNHP and LL, HNHP and HL, LNHP, HNLP
and LNLP conditions under HL, respectively (Tukey post hoc test, all $p > 0.1$) (Fig.
4a,c,f). This demonstrated that increasing temperature within the test range had no
significant effect on POC quota. The combined effects of increasing $p\mathrm{CO_2}$ and
temperature on POC quotas were nutrient dependent. Compared to low $p\mathrm{CO_2}$ and low
temperature (LCLT, Fig. 4a), POC quotas at high $p\mathrm{CO_2}$ and high temperature (HCHT,
Fig. 4d) did not show significant differences in HNHP and LL ($p = 0.79$), in HNHP
and HL ($p = 0.99$), and in LNHP and HL ($p = 0.99$), but increased by 52.2% ± 20.6%
in HNLP and by 45.6% ± 14.8% in LNLP conditions under HL (Tukey post hoc test,
both $p < 0.01$) (Fig. 4a,d,g; Tables 2 and 4). These data showed that high $p\mathrm{CO_2}$ and





low phosphate concentrations enhanced POC quotas of *E. huxleyi*, and that their
combined effects were partly reduced by rising temperature.
The effects of nutrient reduction on POC quota are nutrient specific (Fig. 4).
Compared to HNHP and HL, POC quotas in LNHP did not show a significant
difference (all $p > 0.1$ at LCLT, HCLT, LCHT and HCHT) (Fig. 4a–d,h; Tables 2 and
4). At LC, POC quotas did not significantly differ between HNHP, HNLP and LNLP
conditions (Tukey post hoc test, all $p > 0.1$) (Fig. 4a,c,i,j). In contrast, in HC, they
were 43.3–78.2% larger in HNLP or LNLP than in HNHP (all $p < 0.01$) (Fig. 4b,d,i,j;
Table 2).

**3.4 PIC quota**
Cellular PIC quotas were significantly larger in the future scenario with high levels of
$pCO_2$, temperature and light along with low nutrients concentrations, than PIC quotas
in the present scenario with low levels of $pCO_2$, temperature and light along with
relatively high nutrients concentrations (one-way ANOVA, F = 63.6, $p < 0.01$) (Figs.
2c and 5a,d). The effect of increasing $pCO_2$ on PIC quota is negative, regardless of
presence of other drivers. By comparing PIC quota in all of the HC regimes with their
paired LC regimes (Figs. 5a,b,e and S4), the effects of elevated $pCO_2$ level are clear,
though the extent of reduction in PIC quota depends on which other stressors are
present. Compared to present atmospheric $pCO_2$ levels (LC, Fig. 5a), PIC quotas
under ocean acidification (Fig. 5b) are reduced by 31.8% $\pm 17.1$% in HNHP and LL,
by 34.3% $\pm 10.0$% in HNHP and HL, by 25.0% $\pm 3.8$% in LNHP, by 22.8% $\pm 6.3$% in
HNLP and by 44.6% $\pm 0.9$% in LNLP conditions under HL, respectively (Tukey post
hoc test, all $p < 0.05$) (Fig. 5a,b,e; Tables 2–4). The extent of reduction in PIC quota
is larger under LNLP conditions.





The effects of rising temperature on PIC quota were nutrient dependent, and can be
seen by comparing PIC quotas in the HT regimes with those in their paired LT
regimes (Figs. 5a,c,f and S4). Compared to low temperature (LT, Fig. 5a), PIC quotas
at high temperature (HT, Fig. 5c) did not show significant differences in HNHP and
LL, in HNHP and HL, in LNHP, and in HNLP conditions (Tukey post hoc test, all $p >$
0.05), whereas they decreased by 27.9% $\pm$ 8.4% in LNLP conditions under HL
(Tukey post hoc test, $p < 0.01$) (Fig. 5a,c,f; Tables 2–4). The combined effects of
rising $pCO_2$ and temperature on PIC quota are negative, regardless of which other
drivers are present (Fig. 5a,d,g). Compared to low $pCO_2$ and low temperature (LCLT,
Fig. 5a), PIC quotas in high $pCO_2$ and high temperature (HCHT, Fig. 5d) declined by
11.1% $\pm 10.9\%$ in HNHP and LL ($p = 0.96$), by 32.5% $\pm 2.4\%$ in HNHP and HL ($p <$
0.01), by 42.2% $\pm 3.2\%$ in LNHP ($p < 0.01$), by 10.2% $\pm 7.7\%$ in HNLP ($p = 0.92$),
and by 45.3% $\pm 5.9\%$ in LNLP conditions under HL, respectively ($p < 0.01$) (Fig.
5a,d,g; Table 2).
Effects of both nitrate and phosphate reduction on PIC quota are positive,
regardless of levels of $pCO_2$ and temperature for the range used here (Fig. 5h,i,j).
Compared to HNHP and HL, PIC quotas were larger in LNHP (Tukey post hoc test, $p$
$< 0.01$ in LCLT, HCLT and LCHT conditions; $p = 0.73$ at HCHT condition) (Fig. 5h),
in HNLP, and in LNLP conditions, respectively (all $p < 0.01$ at LCLT, HCLT, LCHT
and HCHT conditions) (Fig. 5a–d,i,j; Table 2). In addition, PIC quotas were larger in
LNLP than in HNLP conditions (Tukey post hoc test, $p < 0.01$ in LCLT and HCLT
conditions; $p = 0.06$ in LCHT; $p = 0.21$ in HCHT conditions) (Fig. 5a–d,i,j). These
data showed that low nitrate and phosphate concentrations act synergistically to
increase PIC quotas, which was moderated under the high $pCO_2$.



### 3.5 PIC / POC value

The ratio of PIC to POC (PIC/POC value) was not significantly different between the future scenario (HCHT HL LNLP) and the current scenario (LCLT LL HNHP) (one-way ANOVA, F = 0.3, $p$ = 0.60) (Figs. 2d and 6a,d). The PIC / POC value followed the same trend as for PIC quotas described above. The effect of increasing $p$CO$_2$ on PIC / POC value was negative, regardless of which other drivers were present (Figs. 6a,b,e and S4), but the extent of reduction in PIC / POC value depended on presence of other drivers. Compared to current atmospheric $p$CO$_2$ levels (LC, Fig. 6a), PIC / POC values under ocean acidification (HC, Fig. 6b) decreased by 50.7% ± 18.2% in HNHP and LL, by 41.8% ± 15.4% in HNHP and HL, by 43.9% ± 5.8% in LNHP, by 63.0% ± 4.2% in HNLP, and by 70.7% ± 2.0% in LNLP conditions under HL, respectively (Tukey post hoc test, all $p$ < 0.05) (Fig. 6a,b; Table 2).

The effect of rising temperature on PIC / POC value was nutrient dependant (Figs. 6a,c,f and S4). Compared to low temperature (LT, Fig. 6a), PIC / POC values at high temperature (HT, Fig. 6c) did not show significant differences in HNHP and LL, in HNHP and HL, in LNHP, and in LNLP conditions (Tukey post hoc test, all $p$ > 0.1), whereas they increased by 39.0% ± 8.9% in HNLP conditions (Tukey post hoc test, $p$ = 0.006) (Fig. 6a,c,f; Table 2). The combined effects of elevated $p$CO$_2$ and temperature on PIC / POC values were negative (Fig. 6a,d,g). Relative to low $p$CO$_2$ and low temperature (LCLT, Fig. 6a), PIC / POC values at high $p$CO$_2$ and high temperature (HCHT, Fig. 6d) did not show significant differences in HNHP and LL, and in HNHP and HL conditions (Tukey post hoc test, both $p$ > 0.1), but they decreased by 39.9% ± 3.0% in LNHP, by 40.6% ± 5.8% in HNLP, and by 67.8% ± 3.1% in LNLP conditions under HL, respectively (Tukey post hoc test, all $p$ < 0.01) (Fig. 6a,d,g; Table 2).





Across all LNHP/HNHP (low/high nitrate) regime pairs, PIC / POC values were
higher in the LNHP regime (Fig. 6h), though the extent of increase in PIC / POC
values depended on $p$CO$_2$ or temperature levels. Compared to HNHP and HL, PIC /
POC values in LNHP were about 106.0% ± 13.0% larger (Tukey post hoc test, $p <$
0.05 in LCLT and LCHT conditions; $p > 0.05$ in HCLT and HCHT conditions) (Fig.
6a–d, h; Table 2). The effect of phosphate on PIC / POC value also depended on
$p$CO$_2$ levels (Fig. 6i). In LC, PIC / POC values were larger in HNLP than in HNHP ($p$
= 0.22 at LCLT; $p < 0.05$ at LCHT conditions), and in LNLP than in LP ($p < 0.01$ at
LCLT; $p = 0.09$ in LCHT conditions) (Fig. 6a,c). In HC conditions, PIC / POC values
did not show significant differences among HNHP, HNLP and LNLP conditions
(Tukey post hoc test, all $p > 0.05$ in HCLT and HCHT conditions) (Fig. 6b,d; Table 2).

**4 Discussion**
Understanding effects of multiple drivers is helpful for improving how
coccolithorphores are represented in models (Krumhardt et al., 2017). Responses of
growth, POC and PIC quotas to ocean acidification have been shown to be modulated
by temperature (Gafar and Schulz, 2018; Tong et al., 2019), light intensity or light
period (light : dark cycle) (Jin et al., 2017; Bretherton et al., 2019), DIN or DIP
concentrations (Müller et al., 2017), combinations of light intensity and nutrients
availability (Zhang et al., 2019) (Table 5). Following up our previous study (Zhang et
al., 2019), we added temperature as a key driver of 5 drivers (Table S2), and explored
how temperature changes would modulate the combined effects of CO$_2$, light, DIN
and DIP that we previously reported. Our data showed that a future ocean climate
change cluster (increasing CO$_2$, temperature, and light levels along with decreasing
DIN and DIP levels) can lower growth rate with increased POC and PIC quota per



cell (Fig. 2) as a result of plastic responses to the drivers. In contrast, observations of
coccolithophore Chl *a* increased from 1990 to 2014 in the North Atlantic, and rising
$CO_2$ and temperature has been aassociated with accelerated growth of
coccolithophores since 1965 in the North Atlantic (Rivero-Calle et al., 2015;
Krumhardt et al., 2016). Our results from laboratory experiments with multiple
drivers experiment instead predicted a different trend with progressive ocean climate
change, suggesting that some key elements of understanding phytoplankton responses
to changing conditions that would enable researchers to connect laboratory studies
and field observations are missing. It should also be noted that regional responses to
ocean global changes could differ due to chemical and physical environmental
differences and species and strain variability among different oceans or regions
(Blanco-Ameijeiras et al., 2016; Gao et al., 2019), and that this could also explain
discrepancies between experiments and observations.
The decreased availability of nitrate or phosphate individually reduced growth rate
and increased PIC quota, respectively, in this experiment. Furthermore, under LNLP
and high $p$CO$_2$ levels, measured growth rates were significantly lower than the
expected values estimated on the basis of the values in LNHP and HNLP conditions
(Fig. S3a). This indicates synergistic negative effects of LN and LP on growth rate, an
evidence that colimitation of N and P is more severe than that by N or P alone. Here,
the extent of synergy between LN and LP on growth rate was calculated to be
8.6% ±2.8% at low temperature and to be 40.6% ±3.8% at high temperature (Fig. S3a),
suggesting modulating effect of temperature on response of growth rate to nutrient
limitations (Thomas et al., 2017). Similarly, at LNLP and low $p$CO$_2$ level, the
measured PIC quota was significantly larger than the expected value (Fig. S3c),
indicating synergistic positive effects of LN and LP on PIC quota, with the extent of



548 synergy being 31.4% ±3.9% at low temperature. LN and LP did not synergistically act

549 to reduce POC quota.

550  While there were always interactions among stressors, increased temperature itself

551 sped up population growth to a relatively consistent value at high light, regardless of

552 nutrient limitation, with statistically significant but small differences over the different

553 nutrient regimes (Fig. 3f). Rising $p\mathrm{CO_2}$ level not only decreased the absolute values of

554 growth rate, but also reduced the positive effect of high temperature on growth. In

555 addition, elevated $p\mathrm{CO_2}$ also altered patterns of growth responses to changes in light

556 and nutrient levels (Fig. 3e–g). Interestingly, low-pH inhibited growth to lesser extent

557 under the high light than under low light (Fig. 3e; Table 2). One possible explanation

558 for this could be that photosynthesis under the high light regime could generate more

559 energy-conserving compounds, which results in faster $p\mathrm{CO_2}$ removal and counteracts

560 the negative effects of low pH. This interaction between low pH and high light was

561 also observed when *E. huxleyi* was grown under incident sunlight (Jin et al., 2017).

562  Increases in temperature reduced PIC quotas under some conditions (high light

563 (HL), HL-LNHP and HL-LNLP) (Fig. 5f), suggesting that the ratio of N:P is

564 important in modulating calcification under warming. One striking result is the

565 consistent negative effect of high $p\mathrm{CO_2}$ on growth and PIC quota, regardless of other

566 stressors. While $p\mathrm{CO_2}$ levels affected the absolute PIC values, the combination of

567 high $p\mathrm{CO_2}$ and warming did not affect the responses to light and nutrients once the

568 direct reduction in PIC quota due to increased $p\mathrm{CO_2}$ was taken into account (Fig. 5g).

569 It has been documented that PIC quotas of *E. huxleyi* reduced at high $p\mathrm{CO_2}$ due to

570 suppressed calcification (Riebesell and Tortell, 2011). This knowledge has been based

571 on experiments under nutrient-replicate or constant conditions without consideration

572 of multiple drivers. In this work, PIC quota of *E. huxleyi* under OA were raised with




increased light intensity and decreased availability of nutrients (Figs. 2 and 5). These
results are consistent with other studies (Perrin et al., 2016; Jin et al., 2017), which
reported that nutrient limitations enhanced calcification, and increased light levels can
partially counteract the negative effects of OA on calcification. Our data also indicate
that effects of ocean climate change on calcification of *E. huxleyi* are more complex
than previously thought (Meyer and Riebesell, 2015). It is worth noting that the
observed higher POC and PIC quotas under future ocean climate change scenario
could be attributed to cell cycle arrest of a portion of the community (Vaulot et al.,
1987). Decreased availabilities of nitrate and phosphate can extend the G1 phase
where photosynthetic carbon fixation and calcification occurred, and lead to lower
dark respiration which reduces carbon consumption (Vaulot et al., 1987; Müller et al.,
2008; Gao et al., 2018).
Low phosphate concentrations can induce high affinity phosphate uptake in *E.*
*huxleyi* (Riegman et al., 2000; Dyhrman and Palenik, 2003; McKew et al., 2015). This
mechanism enables *E. huxleyi* to take up phosphate efficiently at low $pCO_2$
concentrations, so that no significant difference in growth rate was observed between
HNLP and HNHP treatments (Fig. 3a,c). However, at high $pCO_2$, low phosphate
concentration (HNLP) lowered growth of *E. huxleyi* relative to HNHP (Fig. 3a–d;
Table 2). While the affinity of *E. huxleyi* for phosphate under different $pCO_2$ levels
has not been studied, the extra energetic cost of coping with stress from high $pCO_2$
could limit the energy available for the active uptake of phosphate. In addition, the
activity of alkaline phosphatase, which might work to reuse released organic P,
decreases at low pH (Rouco et al., 2013). Finally, the enlarged cell volume in HC and
HNLP (or LNLP) conditions may further reduce nutrient uptake by cells due to
reduced surface to volume ratios, and lower cell division rates (Fig. S5) (Finkel, 2001).





On the other hand, HNLP also affected expressions of genes related to nitrogen
metabolism due to the tight stoichiometric coupling of nitrogen and phosphate
metabolism (Rokitta et al., 2016). Decreased availability of nitrate further limited
nitrogen metabolism of *E. huxleyi* (Rokitta et al., 2014), which lowered the overall
biosynthetic activity and reduced cellular PON quotas (Fig. S10). These explain the
synergistic inhibitions of low-pH, low-phosphate and low-nitrate on growth of *E.*
*huxleyi* (Fig. 3).
POC quotas were larger at high $p$CO$_2$ than at low $p$CO$_2$ under all treatments (Fig. 4;
Table 2), which could be a combined outcome of increased photosynthetic carbon
fixation (Zondervan et al., 2002; Hoppe et al., 2011; Tong et al., 2019) and reduced
cell division (present work), leading to pronounced increase of POC quotas in the
cells grown under low phosphate (HNLP) and high $p$CO$_2$ (Fig. 4). At HNLP and high
$p$CO$_2$ levels, photosynthetic carbon fixation proceeds whereas cell division rate
decreases (Figs. 3 and 4), so reallocation of newly produced particulate organic
carbon (POC) could be slowed down (Vaulot et al., 1987). In this case, over-synthesis
of cellular organic carbon might be released as dissolved organic carbon (DOC),
which can coagulate to transparent exopolymer particles (TEP) and attach to cells
(Biermann and Engel, 2010; Engel et al., 2015). When cells were filtered on GF/F
filters, any TEP would not have be separated from the cells and would have
contributed to the measured POC quota in this study. However, released organic
compounds should be negligible, since they are usually photorespiration-dependent
(Beardall, 1989; Obata et al., 2013).
Synthesis of RNA is large biochemical sinks for phosphate in *E. huxleyi* and other
primary producers (Dyhrman, 2016). Compared to HNHP conditions, HNLP-grown
cells had only 7.8% of total RNA (Fig. S11). This indicates that decreased availability



of phosphate strongly decreased RNA synthesis, which would consequently extend
the interphase of the cell cycle where calcification occurs (Müller et al., 2008). This
could explain why PIC quotas were enhanced by decreased phosphate availability
(Fig. 5). Similarly, decreased availability of nitrate decreased protein (or PON)
synthesis (Fig. S10), which can also block cells in the interphase of the cell cycle, and
increase the time available for calcification in *E. huxleyi* (Vaulot et al., 1987).
Consistently with this, lower rates of assimilation or organic matter production in *E.*
*huxleyi* in LNHP than in HNHP treatments are consistent with more energy being
reallocated to use for calcification (Nimer and Merrett, 1993; Xu and Gao, 2012).
Large PIC quotas of coccolithophores may facilitate accumulation of calcium
carbonate in the deep ocean and increase the contribution of $CaCO_3$ produced by
coccolithophores to calcareous ooze in the pelagic ocean (Hay, 2004). Due to $CaCO_3$
being more dense than organic carbon, larger PIC quotas may facilitate effective
transport of POC to deep oceans, leading to vertical DIC or $CO_2$ gradients of seawater
(Milliman, 1993; Ziveri et al., 2007). While the effects of global ocean climate
changes on physiological processes of phytoplankton can be complex, our results
promote our understanding on how a cosmopolitan coccolithophore responds to future
ocean environmental changes through plastic trait change. While substantial
evolutionary responses to multiple drivers may help further, our results imply that
decreased phosphate availability along with progressive ocean acidification and
warming in surface ocean may reduce the competitive capability of *E. huxleyi* in
oligotrophic waters.





*Data availability.* The data are available upon request to the corresponding author
(Kunshan Gao).



*Author contributions.* YZ, KG designed the experiment. YZ performed this
experiment. All authors analysed the data, wrote and improved the manuscript.



*Competing interests.* The authors declare that they have no conflict of interest.



*Acknowledgements.* This study was supported by National Natural Science
Foundation of China (41720104005, 41806129, 41721005), and Joint Project of
National Natural Science Foundation of China and Shandong province (No.
U1606404).









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





**Figure Legends**

**Figure 1.** Four "baseline" environments were used where $p$CO$_2$ and temperature (temp) were combined in all pairwise combinations: low $p$CO$_2$ + low temp (LCLT, △ ), high $p$CO$_2$ + low temp (HCLT, ＊), low $p$CO$_2$ + high temp (LCHT, □) and high $p$CO$_2$ + high temp (HCHT, ○). Additional stressors were then added to each of the four "baseline" environments. In step 1, low light (LL) was added. In step 2, high light (HL) was added. HL was then maintained for the rest of the experiment. In step 3, low nitrogen was added and high phosphate levels were restored (LNHP). In step 4, low phosphate was added and high nitrogen levels were restored (HNLP). In step 5, both nitrogen and phosphate were low (LNLP). At each step, we measured cell concentration (**a–e**), medium pH$_T$ value (**f–j**), medium $p$CO$_2$ level (**k–o**), dissolved inorganic nitrogen (DIN) (**p–t**) and phosphate (DIP) (**u–y**) concentrations in the media in the beginning and at the end of the incubations. Respectively, LC and HC represent $p$CO$_2$ levels of about 370 and 960 μatm; LT and HT 16 and 20 $^o$C; LL and HL 60 and 240 μmol photons m$^{-2}$ s$^{-1}$ of photosynthetically active radiation (PAR); HN and LN 24.3 and 7.8 μmol L$^{-1}$ NO$_3^-$ at the beginning of the incubation; HP and LP 1.5 and 0.5 μmol L$^{-1}$ PO$_4^{3-}$ at the beginning of the incubations. The samples were taken in the last day of the cultures in each treatment. The values were indicated as the means ± sd of 4 replicate populations for each treatment.

**Figure 2.** Growth rate (**a**), particulate organic (POC, **b**) and inorganic (PIC, **c**) carbon quotas, PIC / POC value (**d**) and cell volume (**e**) of *Emiliania huxleyi* grown under the present (defined as low levels of $p$CO$_2$, temperature and light along with high levels of nutrients) and the future (defined as higher levels of $p$CO$_2$, temperature, and light along with low levels of nutrients due to ocean acidification, warming and shoaling of





upper mixing layer) scenarios. Data were obtained after cells were acclimated to
experimental conditions for 14–16 generations and means ± sd of 4 replicate
populations. Different letters (a, b) in each panel represent significant differences
between future and present ocean conditions (Tukey Post hoc, $p < 0.05$).

**Figure 3.** Growth rates of *E. huxleyi* grown in LCLT (**a**), HCLT (**b**), LCHT (**c**) and
HCHT (**d**) conditions, and the ratio of growth rate at HC to LC (**e**), HT to LT (**f**),
HCHT to LCLT (**g**), LNHP to HNHP (**h**), HNLP to HNHP (**i**) and LNLP to HNHP (**j**).
Data were obtained after cells were acclimated to experimental conditions for 14–16
generations and means ± sd of 4 replicate populations. Horizontal lines in panels (**e**)–
(**j**) showed the value of 1. Different letters (a, b, c, d) in panels (**a**)–(**d**) represent
significant differences between different nutrient treatments (Tukey Post hoc, $p <$
0.05). Detailed experimental conditions were shown in Figure 1.

**Figure 4.** POC quota of *E. huxleyi* grown in LCLT (**a**), HCLT (**b**), LCHT (**c**) and
HCHT (**d**) conditions, and the ratio of POC quota at HC to LC (**e**), HT to LT (**f**),
HCHT to LCLT (**g**), LNHP to HNHP (**h**), HNLP to HNHP (**i**) and LNLP to HNHP (**j**).
Data were obtained after cells were acclimated to experimental conditions for 14–16
generations and means ± sd of 4 replicate populations. Horizontal lines in panels (**e**)–
(**j**) showed the value of 1. Different letters (a, b) in panels (**a**)–(**d**) represent significant
differences between different nutrient treatments (Tukey Post hoc, $p < 0.05$). Detailed
experimental conditions were shown in Figure 1.

**Figure 5.** PIC quota of *E. huxleyi* grown in LCLT (**a**), HCLT (**b**), LCHT (**c**) and
HCHT (**d**) conditions, and the ratio of PIC quota at HC to LC (**e**), HT to LT (**f**),



HCHT to LCLT (**g**), LNHP to HNHP (**h**), HNLP to HNHP (**i**) and LNLP to HNHP (**j**).
Data were obtained after cells were acclimated to experimental conditions for 14–16
generations and means ± sd of 4 replicate populations. Horizontal lines in panels (**e**)–
(**j**) showed the value of 1. Different letters (a, b, c) in panels (**a**)–(**d**) represent
significant differences between different nutrient treatments (Tukey Post hoc, $p <$
0.05). Detailed experimental conditions were shown in Figure 1.

**Figure 6.** PIC / POC value of *E. huxleyi* grown in LCLT (**a**), HCLT (**b**), LCHT (**c**)
and HCHT (**d**) conditions, and the ratio of (PIC / POC value) at HC to LC (**e**), HT to
LT (**f**), HCHT to LCLT (**g**), LNHP to HNHP (**h**), HNLP to HNHP (**i**) and LNLP to
HNHP (**j**). Data were obtained after cells were acclimated to experimental conditions
for 14–16 generations and means ± sd of 4 replicate populations. Horizontal lines in
panels (**e**)–(**j**) showed the value of 1. Different letters (a, b, c) in panels (**a**)–(**d**)
represent significant differences between different nutrient treatments (Tukey Post
hoc, $p < 0.05$). Detailed experimental conditions were shown in Figure 1.

**Figure S1.** Flow chart of the experimental processes. Detailed experimental
conditions were shown in Figure 1.

**Figure S2.** Representative curves for the time course for cell concentrations of *E.*
*huxleyi* under low $p$CO$_2$ (LC), high (HT) or low (LT) temperatures, and high light
(HL) conditions with varying levels of nutrients: HNHP (**a**), LNHP (**b**), HNLP (**c**) and
LNLP (**d**), respectively. Arrow indicates the day when samples were taken in each
treatment. Data were means ± sd of 4 replicate populations. Detailed experimental
conditions were shown in Figure 1.






**Figure S3.** Comparison of growth rate (**a**), POC quota (**b**) and PIC quota (**c**) between
the expected (calculated) values and the measured values under the LNLP treatments.
Different letters (a, b) in each "baseline" environment (LCLT, HCLT, LCHT or
HCHT) represent significant differences (Tukey Post hoc, $p < 0.05$). Detailed
experimental conditions were shown in Figure 1.

**Figure S4.** Heatmap of the changes in growth rate, POC quota, PIC quota and
PIC:POC in each treatment. Values in the present scenario (LC LT LL HNHP) were
considered as the control. A minus sign indicates the reduction in these parameters.

**Figure S5.** Cell volume of *E. huxleyi* grown in LCLT (**a**), HCLT (**b**), LCHT (**c**) and
HCHT (**d**) conditions, and its correlation with POC quota (**e**) and PIC quota (**f**). Data
were obtained after cells were acclimated to experimental conditions for 14–16
generations and means ± sd of 4 replicate populations in panels (**a**)–(**d**). Each point in
panels (**e**) and (**f**) indicates an individual replicate from all experiment. Different
letters (a, b, c) in panels (**a**)–(**d**) represent significant differences between different
nutrient treatments (Tukey Post hoc, $p < 0.05$).

**Figure S6.** Normalized POC quota of *E. huxleyi* to cell volume in LCLT (**a**), HCLT
(**b**), LCHT (**c**) and HCHT (**d**) conditions. Data were obtained after cells were
acclimated to experimental conditions for 14–16 generations and means ± sd of 4
replicate populations. Different letters (a, b) in each panel represent significant
differences between different nutrient treatments (Tukey Post hoc, $p < 0.05$).



**Figure S7.** Normalized PIC quota of *E. huxleyi* to cell volume in LCLT (**a**), HCLT
(**b**), LCHT (**c**) and HCHT (**d**) conditions. Data were obtained after cells were
acclimated to experimental conditions for 14–16 generations and means ± sd of 4
replicate populations. Different letters (a, b, c) in each panel represent significant
differences between different nutrient treatments (Tukey Post hoc, $p < 0.05$).

**Figure S8.** POC production rate of *E. huxleyi* in LCLT (**a**), HCLT (**b**), LCHT (**c**) and
HCHT (**d**) conditions, and the ratio of POC production rate at HC to LC (**e**), HT to LT
(**f**), HCHT to LCLT (**g**), LNHP to HNHP (**h**), HNLP to HNHP (**i**) and LNLP to
HNHP (**j**). Data were obtained after cells were acclimated to experimental conditions
for 14–16 generations and means ± sd of 4 replicate populations. Horizontal lines in
panels (**e**)–(**j**) showed the value of 1. Different letters (a, b, c) in panels (**a**)–(**d**)
represent significant differences between different nutrient treatments (Tukey Post
hoc, $p < 0.05$).

**Figure S9.** PIC production rate of *E. huxleyi* in LCLT (**a**), HCLT (**b**), LCHT (**c**) and
HCHT (**d**) conditions, and the ratio of PIC production rate at HC to LC (**e**), HT to LT
(**f**), HCHT to LCLT (**g**), LNHP to HNHP (**h**), HNLP to HNHP (**i**) and LNLP to
HNHP (**j**). Data were obtained after cells were acclimated to experimental conditions
for 14–16 generations and means ± sd of 4 replicate populations. Horizontal lines in
panels (**e**)–(**j**) showed the value of 1. Different letters (a, b, c) in panels (**a**)–(**d**)
represent significant differences between different nutrient treatments (Tukey Post
hoc, $p < 0.05$).



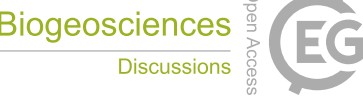

**Figure S10.** PON quota of *E. huxleyi* in LCLT (**a**), HCLT (**b**), LCHT (**c**) and HCHT
(**d**) conditions, and the ratio of PON quota at HC to LC (**e**), HT to LT (**f**), HCHT to
LCLT (**g**), LNHP to HNHP (**h**), HNLP to HNHP (**i**) and LNLP to HNHP (**j**). Data
were obtained after cells were acclimated to experimental conditions for 14–16
generations and means ± sd of 4 replicate populations. Horizontal lines in panels (**e**)–
(**j**) showed the value of 1. Different letters (a, b) in panels (**a**)–(**d**) represent significant
differences between different nutrient treatments (Tukey Post hoc, $p < 0.05$).

**Figure S11.** Normalized RNA quota of *E. huxleyi* to POC quota in HNHP and HNLP
conditions. Data were obtained after cells were acclimated to experimental conditions
for 14–16 generations and means ± sd of 4 replicate populations. Different letters (a, b)
represent significant differences between different nutrient treatments (Tukey Post
hoc, $p < 0.05$).












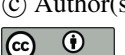



**Table 1.** Carbonate chemistry parameters at the end of the incubation. The values are means ± sd of 4 replicate populations. LL and HL represent 60 and 240 μmol photons m$^{-2}$ s$^{-1}$ of photosynthetically active radiation (PAR), respectively; HN and LN represent 24.3 and 7.8 μmol L$^{-1}$ DIN in the beginning of the incubation; HP and LP represent 1.5 and 0.5 μmol L$^{-1}$ DIP in the beginning of the incubation, respectively.

| | | | $p$CO$_2$ (μatm) | pH (total scale) | TA (μmol L$^{-1}$) | DIC (μmol L$^{-1}$) | HCO$_3^-$ (μmol L$^{-1}$) | CO$_3^{2-}$ (μmol L$^{-1}$) | CO$_2$ (μmol L$^{-1}$) |
|---|---|---|---|---|---|---|---|---|---|
| 16 | LL-HNHP | LC | 371 ±17 | 8.07 ±0.02 | 2266 ±19 | 2017 ±9 | 1823 ±6 | 180 ±8 | 13.4 ±0.6 |
| | | HC | 918 ±21 | 7.73 ±0.02 | 2248 ±45 | 2149 ±39 | 2027 ±35 | 90 ±5 | 33.3 ±0.7 |
| | HL-HNHP | LC | 387 ±22 | 8.06 ±0.02 | 2297 ±12 | 2050 ±17 | 1857 ±20 | 179 ±6 | 14.0 ±0.8 |
| | | HC | 972 ±11 | 7.71 ±0.01 | 2283 ±34 | 2189 ±31 | 2066 ±29 | 88 ±3 | 35.2 ±0.4 |
| | HL-LNHP | LC | 393 ±20 | 8.05 ±0.02 | 2273 ±9 | 2033 ±3 | 1845 ±9 | 174 ±7 | 14.3 ±0.7 |
| | | HC | 1012 ±13 | 7.69 ±0.01 | 2263 ±28 | 2177 ±25 | 2057 ±24 | 84 ±2 | 36.7 ±0.5 |
| | HL-HNLP | LC | 395 ±19 | 8.06 ±0.02 | 2318 ±5 | 2073 ±12 | 1879 ±16 | 179 ±6 | 14.3 ±0.7 |
| | | HC | 958 ±63 | 7.70 ±0.01 | 2205 ±69 | 2117 ±71 | 1999 ±69 | 84 ±1 | 34.7 ±2.3 |
| | HL-LNLP | LC | 375 ±24 | 8.06 ±0.01 | 2181 ±78 | 1947 ±77 | 1767 ±73 | 167 ±3 | 13.6 ±0.9 |
| | | HC | 1014 ±46 | 7.68 ±0.01 | 2198 ±73 | 2118 ±73 | 2002 ±69 | 79 ±2 | 36.7 ±1.7 |
| 20 | LL-HNHP | LC | 349 ±16 | 8.09 ±0.02 | 2257 ±14 | 1963 ±4 | 1741 ±6 | 210 ±8 | 11.3 ±0.5 |
| | | HC | 899 ±40 | 7.74 ±0.02 | 2257 ±53 | 2130 ±45 | 1994 ±40 | 107 ±7 | 29.0 ±1.3 |
| | HL-HNHP | LC | 363 ±11 | 8.08 ±0.01 | 2281 ±16 | 1990 ±18 | 1770 ±19 | 208 ±2 | 11.7 ±0.3 |
| | | HC | 947 ±24 | 7.72 ±0.01 | 2248 ±21 | 2130 ±19 | 1998 ±18 | 102 ±3 | 30.6 ±0.8 |
| | HL-LNHP | LC | 362 ±18 | 8.08 ±0.02 | 2262 ±12 | 1973 ±13 | 1756 ±16 | 206 ±7 | 11.7 ±0.6 |
| | | HC | 970 ±10 | 7.71 ±0.01 | 2271 ±31 | 2155 ±28 | 2021 ±25 | 102 ±3 | 31.4 ±0.3 |
| | HL-HNLP | LC | 370 ±14 | 8.08 ±0.01 | 2314 ±3 | 2023 ±10 | 1800 ±14 | 211 ±4 | 12.0 ±0.4 |
| | | HC | 946 ±47 | 7.71 ±0.01 | 2200 ±72 | 2088 ±72 | 1960 ±68 | 98 ±2 | 30.6 ±1.5 |
| | HL-LNLP | LC | 350 ±18 | 8.08 ±0.01 | 2193 ±71 | 1912 ±68 | 1701 ±63 | 200 ±5 | 11.3 ±0.6 |
| | | HC | 977 ±59 | 7.70 ±0.01 | 2192 ±78 | 2086 ±79 | 1959 ±76 | 95 ±2 | 31.6 ±1.9 |



**Table 2.** Final nitrate and phosphate concentrations (N : P, $\mu$mol L$^{-1}$), growth rate (d$^{-1}$), POC and PIC quotas (pg C cell$^{-1}$), and PIC / POC value. Values in the brackets represent final DIN and DIP concentrations, and standard deviation of 4 replicate populations for growth rate, POC and PIC quotas, and PIC / POC value. Detailed information was shown in Table 1.

| $p$CO$_2$ | T | Light | Final N : P | Growth rate | POC quota | PIC quota | PIC/POC |
|---|---|---|---|---|---|---|---|
| LC | LT | LL | HNHP (17.1 : 0.7) | 0.96 (0.012) | 1.80 (0.14) | 0.38 (0.09) | 0.21 (0.07) |
| | | HL | HNHP (17.3 : 0.5) | 1.09 (0.006) | 2.50 (0.28) | 0.62 (0.05) | 0.25 (0.05) |
| | | HL | LNHP (2.5 : 0.6) | 1.00 (0.013) | 2.07 (0.25) | 0.90 (0.02) | 0.44 (0.05) |
| | | HL | HNLP (15.4 : 0.1) | 1.08 (0.006) | 2.42 (0.08) | 0.83 (0.04) | 0.34 (0.01) |
| | | HL | LNLP (2.4 : 0.1) | 0.99 (0.003) | 2.62 (0.25) | 1.62 (0.14) | 0.63 (0.11) |
| HC | LT | LL | HNHP (18.6 : 0.9) | 0.79 (0.012) | 2.52 (0.33) | 0.26 (0.06) | 0.10 (0.04) |
| | | HL | HNHP (18.2 : 0.5) | 1.04 (0.012) | 2.85 (0.36) | 0.41 (0.06) | 0.15 (0.04) |
| | | HL | LNHP (2.0 : 0.6) | 0.92 (0.026) | 2.75 (0.23) | 0.68 (0.03) | 0.25 (0.03) |
| | | HL | HNLP (15.5 : 0.1) | 0.85 (0.002) | 5.06 (0.34) | 0.64 (0.05) | 0.13 (0.01) |
| | | HL | LNLP (2.7 : 0.1) | 0.67 (0.005) | 4.91 (0.28) | 0.90 (0.01) | 0.18 (0.01) |
| LC | HT | LL | HNHP (16.6 : 0.3) | 1.03 (0.006) | 1.58 (0.11) | 0.43 (0.02) | 0.27 (0.01) |
| | | HL | HNHP (17.3 : 0.3) | 1.46 (0.004) | 2.15 (0.28) | 0.52 (0.07) | 0.25 (0.06) |
| | | HL | LNHP (2.1 : 0.5) | 1.42 (0.004) | 1.68 (0.05) | 0.79 (0.04) | 0.47 (0.03) |
| | | HL | HNLP (17.0 : 0.1) | 1.44 (0.004) | 2.09 (0.03) | 1.00 (0.05) | 0.48 (0.03) |
| | | HL | LNLP (2.1 : 0.1) | 1.39 (0.038) | 2.02 (0.05) | 1.17 (0.13) | 0.58 (0.07) |
| HC | HT | LL | HNHP (16.7 : 0.4) | 0.99 (0.008) | 1.54 (0.12) | 0.34 (0.05) | 0.22 (0.04) |
| | | HL | HNHP (17.9 : 0.5) | 1.43 (0.001) | 2.57 (0.06) | 0.42 (0.02) | 0.16 (0.01) |
| | | HL | LNHP (2.4 : 0.6) | 1.38 (0.009) | 1.97 (0.03) | 0.52 (0.03) | 0.27 (0.01) |
| | | HL | HNLP (17.1 : 0.1) | 1.27 (0.018) | 3.68 (0.50) | 0.74 (0.06) | 0.20 (0.02) |
| | | HL | LNLP (2.2 : 0.1) | 0.87 (0.022) | 3.81 (0.39) | 0.89 (0.10) | 0.20 (0.04) |





**Table 3.** Results of three-way ANOVAs of the effects of temperature (T), $p\mathrm{CO_2}$ (C)
and light intensity (L) and their interaction on growth rate, POC and PIC quotas, and
PIC / POC value. Significant values were marked in bold.

| | | T | C | L | T×C | T×L | C×L | T×C×L |
|---|---|---|---|---|---|---|---|---|
| Growth rate | F | 20037.5 | 477.4 | 23625.8 | 120.0 | 1550.9 | 34.0 | 86.4 |
| | *p* | **<0.01** | **<0.01** | **<0.01** | **<0.01** | **<0.01** | **<0.01** | **<0.01** |
| POC quota | F | 27.1 | 54.4 | 62.0 | 7.4 | 1.9 | < 0.1 | 6.1 |
| | *p* | **<0.01** | **<0.01** | **<0.01** | **0.01** | 0.18 | 0.83 | **0.02** |
| PIC quota | F | 0.4 | 38.6 | 47.6 | 2.3 | 6.6 | 1.6 | 1.1 |
| | *p* | 0.56 | **<0.01** | **<0.01** | 0.14 | **0.02** | 0.22 | 0.31 |
| PIC / POC value | F | 9.9 | 443.6 | 2.0 | 0.8 | 10.0 | 0.6 | 0.3 |
| | *p* | **<0.01** | **<0.01** | 0.17 | 0.38 | **<0.01** | 0.46 | 0.60 |























**Table 4.** Results of four-way ANOVAs of the effects of temperature (T), $p\mathrm{CO_2}$ (C), dissolved inorganic nitrate (N) and phosphate (P) concentrations and their interaction on growth rate, POC and PIC quotas, and PIC / POC value. Significant values were marked in bold.

| | Growth rate | | POC quota | | PIC quota | | PIC / POC value | |
|---|---|---|---|---|---|---|---|---|
| | F | $p$ | F | $p$ | F | $p$ | F | $p$ |
| T | 500026.0 | **<0.01** | 297.4 | **<0.01** | 30.2 | **<0.01** | 82.8 | **<0.01** |
| C | 5798.0 | **<0.01** | 162.8 | **<0.01** | 376.2 | **<0.01** | 787.3 | **<0.01** |
| N | 4542.0 | **<0.01** | 157.0 | **<0.01** | 84.4 | **<0.01** | 127.6 | **<0.01** |
| P | 5347.0 | **<0.01** | 206.5 | **<0.01** | 474.6 | **<0.01** | 0.1 | 0.74 |
| T×C | 6899.0 | **<0.01** | 52.2 | **<0.01** | 0.2 | 0.68 | 7.2 | **<0.01** |
| T×N | 510.0 | **<0.01** | 5.6 | **0.02** | 60.0 | **<0.01** | 7.9 | **<0.01** |
| T×P | 39.0 | **<0.01** | 5.2 | **0.03** | 9.4 | **<0.01** | 16.2 | **<0.01** |
| C×N | 1265.0 | **<0.01** | 107.2 | **<0.01** | 9.5 | **<0.01** | 3.1 | 0.09 |
| C×P | 1718.0 | **<0.01** | 174.1 | **<0.01** | 14.7 | **<0.01** | 88.0 | **<0.01** |
| N×P | 179.0 | **<0.01** | 19.7 | **<0.01** | 10.7 | **<0.01** | 14.3 | **<0.01** |
| T×C×N | 35.0 | **<0.01** | <0.1 | 0.81 | 0.2 | 0.67 | 1.9 | 0.17 |
| T×C×P | 27.0 | **<0.01** | 5.5 | **0.02** | 0.1 | 0.71 | 1.0 | 0.31 |
| T×N×P | 96.0 | **<0.01** | <0.1 | 0.80 | 15.7 | **<0.01** | 3.3 | 0.08 |
| C×N×P | 241.0 | **<0.01** | 0.4 | 0.56 | 8.2 | **<0.01** | 1.2 | 0.28 |
| T×C×N×P | 105.0 | **<0.01** | 3.9 | 0.05 | 22.4 | **<0.01** | 4.5 | **0.04** |



**Table 5.** List of the physiological responses of *E. huxleyi* to the concurrent changes in
multiple drivers investigated by the laboratory incubations in the published studies. '↑'
represents increase, '↓' represents decrease, and 'n' represents no significant change
to simultaneous changes in multiple drivers. C, T, L, N, P and μ represent $CO_2$ (μatm),
temperature (°C), light intensity (μmol photons $m^{-2}$ $s^{-1}$), dissolved inorganic nitrogen
and phosphate (μmol $L^{-1}$), and growth rate, respectively. Simultaneous changes in
multiple drivers were marked in bold. [1] represents De Bodt et al., (2010), [2]
Borchard et al., (2011), [3] Sett et al., (2014), [4] Gafar and Schulz, (2018), [5] Tong
et al., (2019), [6] Jin et al., (2017), [7] Bretherton et al., (2019), [8] Rost et al., (2002),
[9] Feng et al., (2008), [10] Müller et al. (2012), [11] Perrin et al., (2016), [12]
Leonardos and Geider, (2005), [13] Matthiessen et al., (2012), [14] Zhang et al.,
(2019), [15] this study.

| Strain | C | T | L | N | P | μ | POC | PIC | PIC:POC | Cite |
|---|---|---|---|---|---|---|---|---|---|---|
| AC481 | **380 to 750** | **13 to 18** | 150 | 32 | 1 | n | ↑ | ↓ | ↓ | [1] |
| PML B92/11 | **300 to 900** | **14 to 18** | 300 | 29 | 1 | ↑ | n | ↓ | ↓ | [2] |
| PML B92/11 | **400 to 1000** | **10 to 20** | 150 | 64 | 4 | ↑ | ↑ | ↓ | ↓ | [3] |
| PML B92/11 | **400 to 1000** | **10 to 20** | 150 | 64 | 4 | ↑ | ↓ | ↓ | | [4] |
| PML B92/11 | **400 to 1000** | **15 to 24** | 190 | 100 | 10 | ↑ | ↑ | ↓ | ↓ | [5] |
| CCMP2090 | **395 to 1000** | 20 | **57 to 567** | 110 | 10 | ↑ | ↑ | | | [6] |
| NZEH | **390 to 1000** | 20 | **175 to 300** | 100 | 10 | ↓ | ↑ | ↑ | ↑ | [7] |
| PCC124-3 | **390 to 1000** | 20 | **175 to 300** | 100 | 10 | ↑ | n | ↑ | ↑ | [7] |
| PCC70-3 | **390 to 1000** | 20 | **175 to 300** | 100 | 10 | ↑ | n | ↑ | ↑ | [7] |
| PML B92/11 | **140 to 880** | 15 | **80 to 150** | 100 | 6 | ↑ | ↑ | ↓ | ↓ | [8] |
| PML B92/11 | **395 to 1000** | 20 | **54 to 457** | 110 | 10 | ↑ | ↑ | ↓ | ↓ | [6] |
| PML B92/11 | **400 to 1000** | 20 | **50 to 1200** | 64 | 4 | ↑ | ↑ | ↑ | | [4] |
| RCC962 | **390 to 1000** | 20 | **175 to 300** | 100 | 10 | ↓ | ↑ | n | ↓ | [7] |
| CCMP371 | **375 to 750** | **20 to 24** | **50 to 400** | 100 | 10 | ↑ | n | ↓ | ↓ | [9] |
| B62 | **280 to 1000** | 20 | 300 | **88 to 9** | 4 | | ↑ | ↓ | ↓ | [10] |
| RCC911 | 400 | 20 | **30 to 140** | **100 to 5** | 6 | ↑ | ↑ | ↑ | ↑ | [11] |
| RCC911 | 400 | 20 | **30 to 140** | 100 | **6 to 0.6** | ↑ | ↑ | ↑ | ↑ | [11] |





| | | | | | | | | | | |
|---|---|---|---|---|---|---|---|---|---|---|
| PML92A | **360 to 2000** | 18 | **80 to 500** | 200 | **6.7 to 40** | n | ↑ | | | [12] |
| A | **460 to 1280** | 16 | 130 | **17 to 9** | **0.2 to 0.5** | | ↓ | ↓ | | [13] |
| PML B92/11 | **410 to 920** | 20 | **80 to 480** | **100 to 8** | 10 | ↓ | ↓ | ↑ | ↑ | [14] |
| PML B92/11 | **410 to 920** | 20 | **80 to 480** | 100 | **10 to 0.4** | ↓ | ↑ | n | ↓ | [14] |
| PML B92/11 | **370 to 960** | **16 to 20** | **60 to 240** | **24 to 8** | **1.5 to 0.5** | ↓ | ↑ | ↑ | n | [15] |




























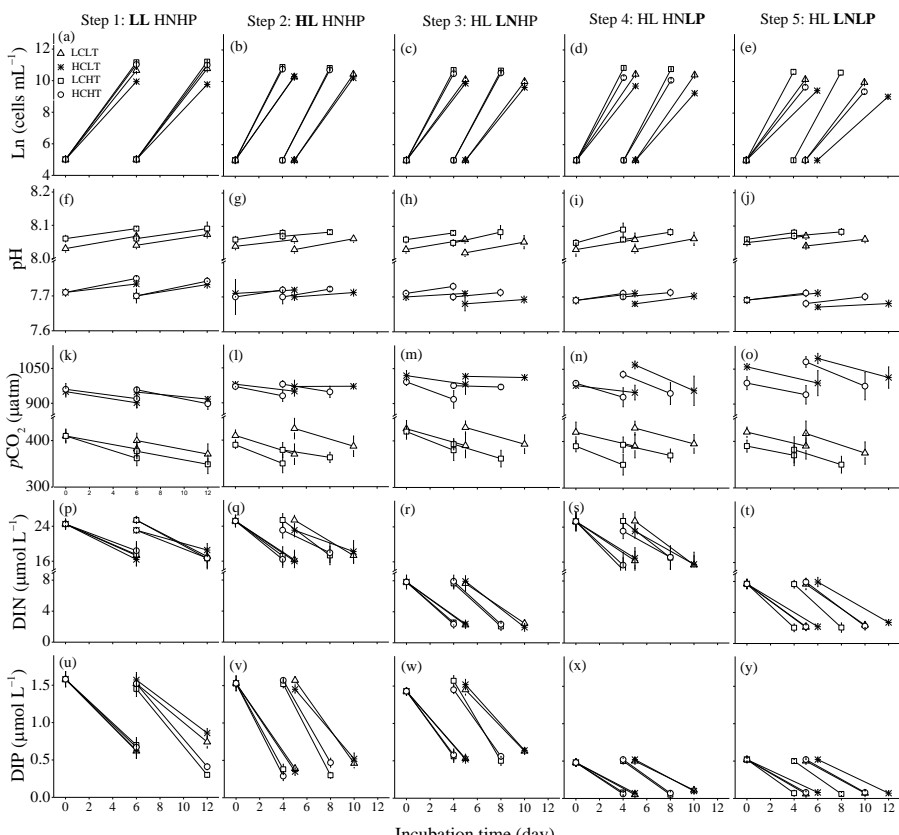




Figure 1












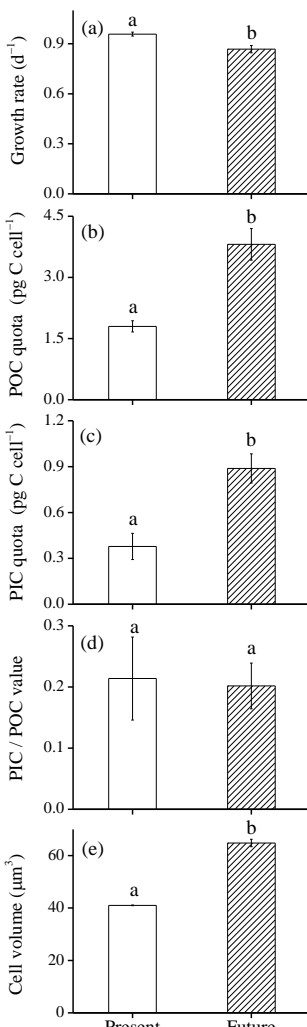



Figure 2





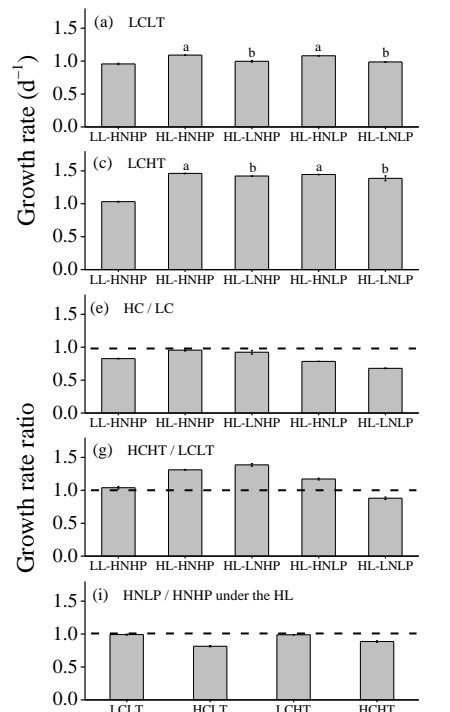
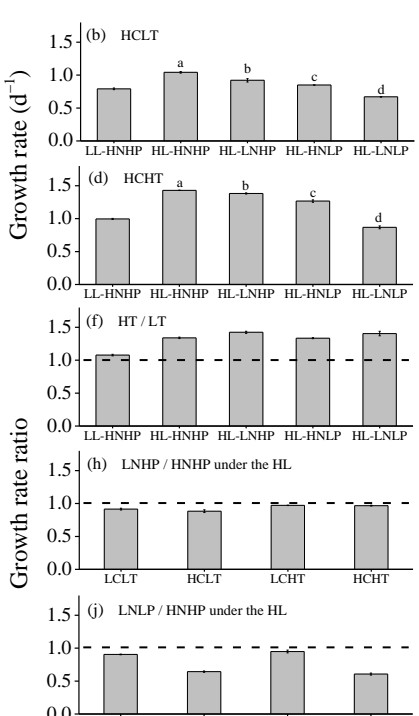




Figure 3














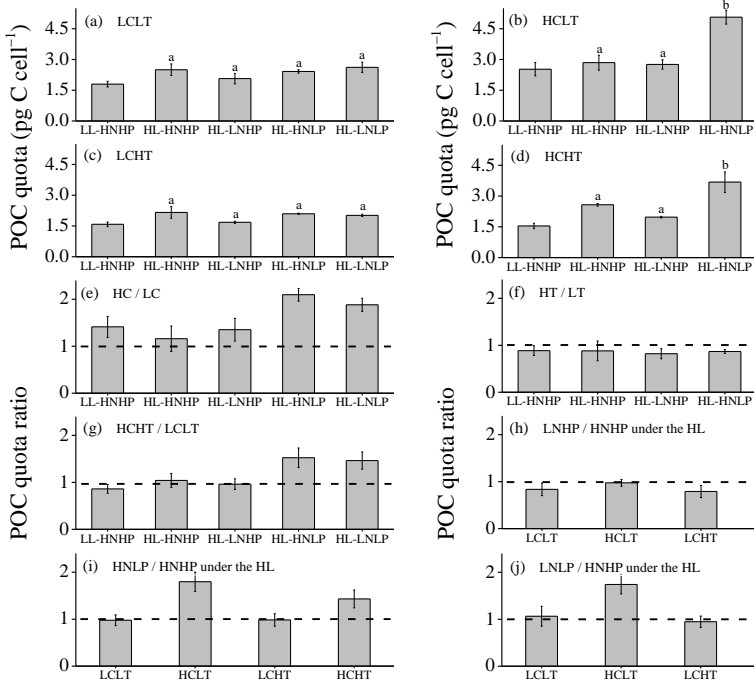





Figure 4












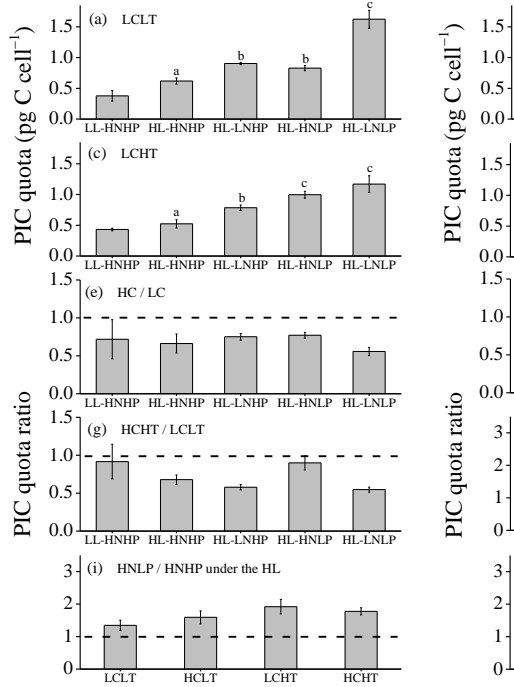
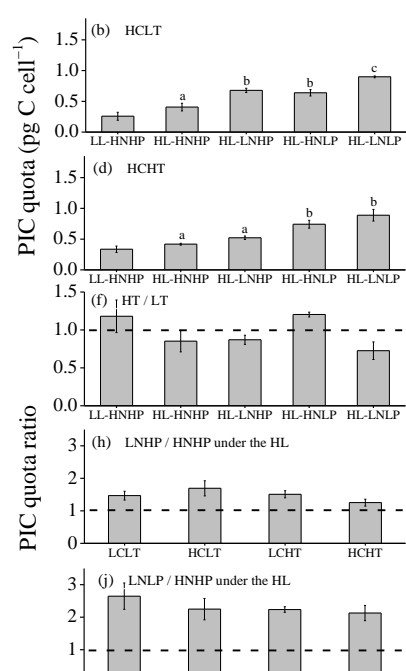





Figure 5














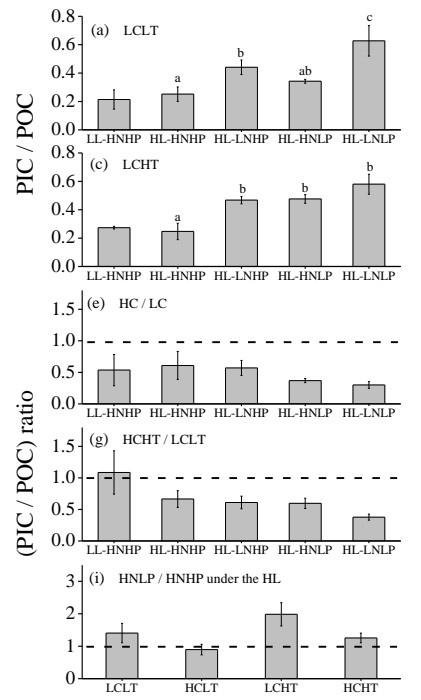
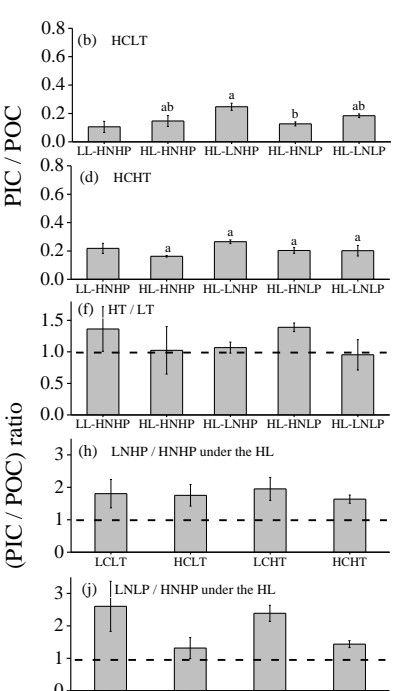





Figure 6
