# Peer review of "Reduced growth with increased quotas of particulate organic and inorganic carbon in the coccolithophore *Emiliania huxleyi* under future ocean climate change conditions"

_Biogeosciences, 2020_

## Referee Comment (RC1) · Anonymous Referee #1 · 18 Aug 2020

General summary:

The Authors explore the effects of future ocean climate change scenarios on Emiliania huxleyi including several important environmental drivers in a "environmental cluster-ing" method that allows for co-variation of environmental drivers that have been iden-tified to affect different trait responses. They found that under future ocean climate change scenarios the growth of the coccolithophore is reduced and biogeochemically relevant traits (i.e. PIC and POC quotas) are increased. However, nutrient limitation (expected in future climate change scenarios) affected these traits in different ways

and interactions. Therefore the Authors conclude, that their study contributes to the understanding of the response of E. huxleyi to future climate change.

General comments:

The manuscript is clearly written and structured. The study is well designed and explained considering how complex experiments tend to get when increasing the number of environmental drivers. Also, the rationale behind using a "environmental clustering" approach is clearly stated and the co-variation of the environmental drivers is well deduced from previous studies and literature. It is interesting to see in this study how much variation in response to nutrient availability and light there is, even though there are clear differences in PIC, POC quota and growth between ambient and future scenarios. These differences are put into context and discussed well in the manuscript. There is however, one critical point that I think is not well discussed and needs to be considered in the discussion. In the beginning of the discussion, the Authors make the comparison between ocean observations of Coccolithophores and try to highlight discrepancies to the lab experiment. While these comparisons are nearly impossible, because laboratory conditions are so different from natural conditions, I feel that this point merits further discussion. Environmental variation due to different geographical regions affects the environmental history an organism has experienced and thus how an organism can and will react to changes in the environment. In addition within species and within functional group variation in plastic responses of growth and other traits is well established (other publications by Zhang et al have already shown this as well) and can affect how "a species" responds to environmental change. Since the Authors want to make a conclusion what their study tells us about how a cosmospolitan and biogeochemically relevant group of phytoplankton react to future ocean scenarios and not only conclude something about the combination and co-variation of environmental drivers, I feel that ecological variability and environmental variability should be taken into account in the discussion. One way to approach this "gap" of knowledge could be a discussion about what experimental conditions (based on the given study)

could now be focused on to further characterize the responses of other coccolithophore strains that i) come from environmentally different regions, ii) are more recently isolated and thus not acclimatized to long times spent in the laboratory.

Specific comments:

Line 120: here the Authors imply, that their study will help understand how biogeo-chemically relevant phytoplankton change in future climate change scenarios but this is not adequately discussed later on (see general comments)

Line 161: "adding low light" is misleading. Would it be possible to say that light was reduced?

Line 151 and 161: it would be good to have an idea about nutrient and light concentrations here already. The information following in line 190 comes a bit late and could even be combined as later on the $pCO_2$ manipulation is in focus.

Line 175: I do not fully follow the rationale behind adding the nutrient limitation stepwise

PIC quota Line 434 ff: I stumbled over the way that the effect of future ocean scenarios are increasing PIC quotas followed by the explanation of how PIC is reduced with increasing $pCO_2$. It would be helpful if there was one more sentence that relates the different results. In addition it could be helpful to highlight in Fig. 2-6, what parts of all of the results are used for the ambient-future comparison. Then the in-between data that are very interesting could become more clear.

Line 531: Please see the general comment: here the discussion should go further because not only oceanic conditions may be different but ecologically within species and functional groups there are many differences that can affect the results.

Line 612 ff.: I see how considering TEP as part of POC quota is important. But then the Authors also say that it is negligible. As it is written currently, the two sentences contradict each other a bit. Consider rephrasing

Line 620ff: consider moving this part of the discussion about RNA and protein metabolism to where cell cycle is already discussed in line 580. Could fit better together.

Line 643: The conclusion about competitive ability comes a bit "sudden". Consider mentioning the implications of nutrient uptake on competitive ability earlier in the discussion where phosphate and nutrient uptake related changes are discussed.

Technical/language comments:

Line 234: "taken" should be "took"

Line 571: should say nutrient-replete

Line 598: On the other hand to what? Please rephrase

Line 620: type: "a" is missing

Fig. 1: Please indicate in the legend that experimental steps were done in a consecutive manner. Also this might be helpful to mention again in Fig S1. Visually this implies that the steps are done in parallel, but in the methodological description they are explained as being done one after another.

---

## Referee Comment (RC2) · Anonymous Referee #2 · 5 Sep 2020

This study examined the interactive effects of multiple environmental drivers, including $CO_2$, temperature, light, dissolved inorganic nitrogen and phosphate, on the physiology of the ecologically important coccolithophore Emiliania huxleyi. The authors used a novel step-wise laboratory manipulation method, in order to compare the single driver effects with the cumulative effects of multiple drivers. The results suggest that the simultaneous manipulation of ocean edification, warming and decreased nutrient (N and P) concentration significantly decreased the growth rate and increased the cellular carbon quotas. These findings will contribute to our understanding of the coccolithophorerelated ecological and biogeochemical responses to future ocean global change.

Overall, the experiment was well designed and the data was carefully analyzed. The manuscript is also well structured and written. However, I have some major comments listed below.

1. The authors refer the manipulated conditions as "future conditions" in the discussion. Therefore, it would be better to justify why these environmental conditions represent the future global change scenario. For example, the irradiance levels and the nutrient concentrations set up for the experiment are not within the ranges listed in Table S1. The physiological response of E. huxleyi would be different under different levels of environmental conditions (i.e. irradiance and nutrient). How will the results of this study be extrapolated to the future global change scenario?

2. The coccolithophore Emiliania huxleyi is a cosmopolitan species. Previous studies have shown strain-specific responses of E. huxleyi to environmental changes (especially ocean acidification). I would suggest the authors to expand the discussion on Table 5 a little further.

Some other specific comments:

Lines 163: "low nitrogen was added…" I don't think this is a correct expression of introducing low nitrate concentration. Could the authors also specify how the nitrate concentration was reduced? The same for line 164, "low phosphate was added..".

Line 269: The cell diameter was measured for the whole coccosphere, with coccoliths attached. However, both PIC quota and PIC/POC ratio was changed by different experimental manipulations, especially by alteration of pCO2. This would have also resulted in changes in coccolith thickness. I was wondering if the authors have considered this when calculating the cell-volume normalized particulate organic elemental quotas.

Lines 527-531: This sentence is too long, please split to two.

Line 556: "low-pH inhibited growth.." Here the authors indicate it was mainly the effects

of pH instead of changing pCO2, please add some explanations on this.

Line 559: Please add a reference after the sentence "photosynthesis under the high light regime could generate more energy-conserving compounds".

Line 561: Please specify the strain of the E. huxleyi examined in Jin et al., as well as in line 569.

Line 575: Why was PIC quota increased under high light? Please add some explanations.

Line 617: The sentence "released organic compounds should be negligible. . ." contradicts to the previous expression of "over-synthesis of cellular organic carbon might be released as dissolved organic carbon.." in lines 612-613.

Fig. 1 Please label the symbols in the graph for a better understanding of the treatments.

Fig. S1 I think it would be better to move this figure to the main manuscript, instead of being in the supplementary materials, in order to make a better understanding of the step-wise experimental design.

Fig. S11 How was the RNA concentration measured? This is not presented in the methods section.

---

## Author Comment (AC2) · 2 Oct 2020

Supplemental Information

**Table S1.** Average surface seawater $pCO_2$ level (µatm), sea surface temperature (ºC), daytime mean irradiance (µmol photons $m^{-2}$ $s^{-1}$), and nutrient concentration (µmol $L^{-1}$) during 2000 to 2007 in Norwegian coastal waters where the *E. huxleyi* strain used here was isolated from (Larsen et al., 2004; Locarnini et al., 2006; Omar et al., 2010), and in projected levels for 2100 in high-latitude province in North Atlantic Ocean (Future) (Boyd et al., 2015).

| | $pCO_2$ (µatm) | Temperature (ºC) | Daily irradiance (µmol photons $m^{-2}$ $s^{-1}$) | Nitrate (µmol $L^{-1}$) | Phosphate (µmol $L^{-1}$) |
|---|---|---|---|---|---|
| 2000–2007 | 240 – 400 | 6.0 – 16.0 | 120 – 350 | 0 – 7.0 | 0.1 – 0.5 |
| Future | 580 – 970 | 7.9 – 19.0 | 156 – 455 | 0 – 4.9 | 0.1 – 0.3 |

**Table S2.** Comparison of experiment treatments between the studies of Zhang et al. (2019) and this work. Main differences between two studies were marked in bold.

| Driver | The study of Zhang et al. (2019) | | | The present study | | |
|---|---|---|---|---|---|---|
| $p\mathrm{CO_2}$ (µatm) | 410, 920 | | | 370, 960 | | |
| Temperature (ºC) | 20 | | | **16**, 20 | | |
| Light (µmol photons m$^{-2}$ s$^{-1}$) | **80, 120, 200, 320, 480** | | | 60, 240 | | |
| DIN (µmol L$^{-1}$) | **100**, 8 | | | **24**, 8 | | |
| DIP (µmol L$^{-1}$) | **10**, 0.4 | | | **1.5**, 0.5 | | |
| Experimental setup | HNHP | LC | **5 light levels** | LC | LT | **LL-HNHP** |
| | | HC | | | HT | **HL-HNHP** |
| | LNHP | LC | | | | **HL-LNHP** |
| | | HC | | HC | LT | **HL-HNLP** |
| | HNLP | LC | | | HT | **HL-LNLP** |
| | | HC | | | | |

(a)

[Figure]

**seawater**

**16ºC (LT)**
LC   HC

**20ºC (HT)**
LC   HC

**LC + LT**   **HC + LT**

**LC + HT**   **HC + HT**

**310 ml seawater**          **310 ml seawater**

(b)

LC LT   HC LT
LC HT   HC HT

**Step 1**   **Step 2**   **Step 3**   **Step 4**   **Step 5**

**Low Light** + HNHP

**High Light** + HNHP

High Light + **LNHP**

High Light + HNLP

High Light + **LNLP**

*Emiliania huxleyi* was grown in each experimental condition for 14 –16 generations, and then growth rate, POC and PIC quotas were measured

Figure S1

[Figure]

Figure S2

[Figure]

Figure S3

[Figure]

Figure S4

[Figure]

Figure S5

[Figure]

Figure S6

[Figure]

Figure S7

[Figure]

Figure S8

[Figure]

Figure S9

[Figure]

Figure S10

[Figure]

Figure S11

**References**

Boyd, P. W., Lennartz, S. T., Glover, D. M., and Doney, S. C.: Biological ramifications of climate-change-mediated oceanic multi-stressors, Nat. Clim. Change, 5, 71–79, doi: 10.1038/nclimate2441, 2015.

Larsen, A., Flaten, G. A. F., Sandaa, R., Castberg, T., Thyrhaug, R., Erga, S. R., Jacquet, S., and Bratbak, G.: Spring phytoplankton bloom dynamics in Norwegian coastal waters: Microbial community succession and diversity, Limnol. Oceanogr., 49, 180–190, doi: 10.4319/lo.2004.49.1.0180, 2004.

Locarnini, R. A., Mishonov, A. V., Antonov, J. I., Boyer, T. P., and Garcia, H. E.: World ocean atlas 2005, V. 1: Temperature, edited by: Levitus, S., NOAA Atlas NESDIS 61. U. S. Government Printing Office, 123–134, 2006.

Omar, A. M., Olsen, A., Johannessen, T., Hoppema, M., Thomas, H., and Borges, A. V.: Spatiotemporal variations of $f$CO$_2$ in the North Sea, Ocean Sci., 6, 77–89, doi: 10.5194/osd-6-1655-2009, 2010.

Zhang, Y., Fu, F., Hutchins, D. A., and Gao. K.: Combined effects of CO$_2$ level, light intensity and nutrient availability on the coccolithophore *Emiliania huxleyi*, Hydrobiologia, 842, 127–141, doi: 10.1007/s10750-019-04031-0, 2019.

---

## Author Comment (AC3) · 2 Oct 2020

[revised manuscript text omitted]

Figure 1

[Figure]

Figure 2

[Figure]

[Figure]

Figure 3

[Figure]

Figure 4

[Figure]

[Figure]

Figure 5

[Figure]

Figure 6

---

## Author Response (AR1)

Dear Editor,

We thank the referees for their supportive and constructive comments to our manuscript. We have responded to the comments point by point as follows. The revised sentences or contents are underlined.

Responses to comment 1 are following:

General comments:

The manuscript is clearly written and structured. The study is well designed and explained considering how complex experiments tend to get when increasing the number of environmental drivers. Also, the rationale behind using a "environmental clustering" approach is clearly stated and the co-variation of the environmental drivers is well deduced from previous studies and literature. It is interesting to see in this study how much variation in response to nutrient availability and light there is, even though there are clear differences in PIC, POC quota and growth between ambient and future scenarios. These differences are put into context and discussed well in the manuscript. There is however, one critical point that I think is not well discussed and needs to be considered in the discussion. In the beginning of the discussion, the Authors make the comparison between ocean observations of Coccolithophores and try to highlight discrepancies to the lab experiment. While these comparisons are nearly impossible, because laboratory conditions are so different from natural conditions, I feel that this point merits further discussion. Environmental variation due to different geographical regions affects the environmental history an organism has experienced and thus how an organism can and will react to changes in the environment. In addition within species and within functional group variation in plastic responses of growth and other traits is well established (other publications by Zhang et al have already shown this as well) and can affect how "a species" responds to environmental change. Since the Authors want to make a conclusion what their study tells us about how a cosmospolitan and biogeochemically relevant group of phytoplankton react to future ocean scenarios and not only conclude something about the combination and co-variation of environmental drivers, I feel that ecological variability and environmental variability should be taken into account in the discussion.

One way to approach this "gap" of knowledge could be a discussion about what experimental conditions (based on the given study) could now be focused on to further characterize the responses of other coccolithophore strains that i) come from environmentally different regions, ii) are more recently isolated and thus not acclimatized to long times spent in the laboratory.

Response: We agree with the suggestions from the referee, and have revised the discussion and added related analysis on this aspect with further references to extra literatures at lines 545–560 in the marked-up manuscript version (below): 'Different *E. huxleyi* strains displayed optimal responses to a broad range of temperature or $CO_2$ level, and *E. huxleyi* strains isolated from different regions showed local adaptation to temperature or $CO_2$ level (Zhang et al., 2014; 2018). Strain-specific responses of growth, POC and PIC production rates in *E. huxleyi* isolated from different regions to changing seawater carbonate chemistry have also been documented (Langer et al., 2009). It has been suggested that inter-strain genetic variability has greater potential to induce larger phenotypic differences than the phenotypic plasticity of a single strain cultured under a broad range of variable environmental conditions (Blanco-Ameijeiras et al., 2016). On the other hand, the genetic adaptation to culture experimental conditions over time may no longer accurately represent the cells in the sea, as reflected in a diatom (Guan and Gao, 2008). Phytoplankton species that had been maintained under laboratory conditions might have lost original traits and display different responses to environmental changes (Lakeman et al., 2009). The strain used in this study has been kept in the laboratory for about 30 years, and the data obtained in this work can hardly reflect relation to its biogeographic origin.'

Specific comments:
Line 120: here the Authors imply, that their study will help understand how biogeochemically relevant phytoplankton change in future climate change scenarios but this is not adequately discussed later on (see general comments).

Reponse: We have revised this part as indicated at lines 537–540 in the marked-up manuscript version (below): 'We have to admit that results from laboratory experiments can hardly extrapolate to natural conditions. Nevertheless, our data provide mechanistic understanding of the combined effects of ocean climate change drivers, which can be useful in analyzing field observations.'

Line 161: "adding low light" is misleading. Would it be possible to say that light was reduced?

Response: Agree. We have reworded the words, and changed 'added' to 'supplied' at lines 164 –167 in the marked-up manuscript version (below).

Line 151 and 161: it would be good to have an idea about nutrient and light concentrations here already. The information following in line 190 comes a bit late and could even be combined as later on the pCO2 manipulation is in focus.

Response: We added one sentence: 'Initial DIN and DIP concentration were 24 μmol $L^{-1}$ and 1.5 μmol $L^{-1}$, respectively, and initial light intensity was 60 μmol photons $m^{-2}$ $s^{-1}$.' in lines 154–155 in the marked-up manuscript version (below).

Line 175: I do not fully follow the rationale behind adding the nutrient limitation stepwise

Response: we added "Such stepwise reduction of nutrients levels would be useful for us to analyze effects of nitrate and phosphate separately, and be expected to have implications for the cells episodically exposed to different levels of nutrients in the sea." in lines 180–183 in the marked-up manuscript version (below).

PIC quota Line 434 ff: I stumbled over the way that the effect of future ocean scenarios are increasing PIC quotas followed by the explanation of how PIC is reduced with increasing pCO2. It would be helpful if there was one more sentence that relates the different results. In addition it could be helpful to highlight in Fig. 2-6, what parts of all of the results are used for the ambient-future comparison. Then the in-between data that are very interesting could become more clear.

Response: We added one sentence: 'However, the opposite results were found under the elevated $CO_2$ treatment alone.' in lines 443–444 in the marked-up manuscript version (below). We added these contents 'The results shown in the black column were used for the ambient-future comparison in figure 2' in figure legends of figures 3–6 (lines 1056–1057, 1065–1067, 1076–1077, 1086–1087 in the marked-up manuscript version (below)).

Line 531: Please see the general comment: here the discussion should go further because not only oceanic conditions may be different but ecologically within species and functional groups there are many differences that can affect the results.

Response: We agree with the suggestions of this referee. Please see the response to general comment 1, we have revised the discussion and added related analysis on this aspect with further references to extra literatures (lines 545–560).

Line 612 ff.: I see how considering TEP as part of POC quota is important. But then the Authors also say that it is negligible. As it is written currently, the two sentences contradict each other a bit. Consider rephrasing.

Response: We have deleted these contents 'However, released organic compounds should be negligible, since they are usually photorespiration-dependent (Beardall, 1989; Obata et al., 2013)' after lines 666–667 in the marked-up manuscript version (below)).

Line 620ff: consider moving this part of the discussion about RNA and protein metabolism to where cell cycle is already discussed in line 580. Could fit better together.

Response: Agree. We have moved the contents of the discussion about RNA and protein metabolism to where cell cycle is discussed in lines 616–629 in the marked-up manuscript version (below).

Line 643: The conclusion about competitive ability comes a bit "sudden". Consider mentioning the implications of nutrient uptake on competitive ability earlier in the discussion where phosphate and nutrient uptake related changes are discussed.

Response: Agree. We have move the contents 'While substantial evolutionary responses to multiple drivers may help further, our results imply that decreased phosphate availability along with progressive ocean acidification and warming in surface ocean may reduce the competitive capability of *E. huxleyi* in oligotrophic waters.' to where phosphate uptake is discussed in lines 643–646 in the marked-up manuscript version (below).

Technical/language comments:

Line 234: "taken" should be "took"

Response: Thanks. 'taken' is changed to 'took' in line 239.

Line 571: should say nutrient-replete

Response: Thanks. 'nutrient-replicate' is changed to 'nutrient-replete' in line 600.

Line 598: On the other hand to what? Please rephrase

Response: 'On the other hand' is changed to 'Meanwhile' in line 646.

Line 620: type: "a" is missing

Response: Thanks. 'a' is added in line 616.

Fig. 1: Please indicate in the legend that experimental steps were done in a consecutive manner. Also this might be helpful to mention again in Fig S1. Visually this implies that the steps are done in parallel, but in the methodological description they are explained as being done one after another.

Response: Thanks. We added these contents 'Experimental steps were done in a consecutive manner.' in line 1028, and in lines 1090–1091.

Responses to comment 2 are following:

1. The authors refer the manipulated conditions as "future conditions" in the discussion. Therefore, it would be better to justify why these environmental conditions represent the future global change scenario. For example, the irradiance levels and the nutrient concentrations set up for the experiment are not within the ranges listed in Table S1. The physiological response of *E. huxleyi* would be different under different levels of environmental conditions (i.e. irradiance and nutrient). How will the results of this study be extrapolated to the future global change scenario?

Response: Under the LNLP condition, initial DIN concentration was 8 μmol L$^{-1}$ and initial DIP concentration was about 0.5 μmol L$^{-1}$. During the incubation, DIN and DIP concentrations reduced to about 2.7 μmol L$^{-1}$ and 0.1 μmol L$^{-1}$, respectively, at the end of the incubation (Table 2). DIN and DIP were 0–4.9 μmol L$^{-1}$ and 0.1–0.3 μmol L$^{-1}$, respectively, under the future conditions (Table S1). So, nutrient concentrations were within the ranges listed in Table S1 at the end of the incubation where cell concentration, cellular POC and PIC quotas were measured. In addition, high light intensity was 240 μmol photons m$^{-2}$ s$^{-1}$ during the cultures, and was also within the ranges of irradiance under the future conditions where irradiance was 156–455 μmol photons m$^{-2}$ s$^{-1}$ (Table S1).

We agree that the physiological response of *E. huxleyi* would be different under different levels of environmental conditions. 'We have to admit that results from laboratory experiments can hardly extrapolate to natural conditions. Nevertheless, our data provide mechanistic understanding of the combined effects of ocean climate change drivers, which can be useful in analyzing field observations.' These contents were added in lines 537–540 in the marked-up manuscript version (below).

2. The coccolithophore *Emiliania huxleyi* is a cosmopolitan species. Previous studies have shown strain-specific responses of *E. huxleyi* to environmental changes (especially ocean acidification). I would suggest the authors to expand the discussion on Table 5 a little further.

Response: As mentioned in response to general comment 1, strain-specific responses in growth rate, POC and PIC production rates of *E. huxleyi* to a range of CO$_2$ or temperature have been reported by Langer et al. (2009) and Zhang et al. (2014; 2018). In addition, Blanco-Ameijeiras et al. (2016) examined variability in cellular contents of POC and PIC, magnesium (Mg) and strontium (Sr) of 13 *E. huxleyi* strains under identical culture conditions. We added related analysis on this aspect with further references to extra literatures in lines 545–560 in the marked-up manuscript version (below).

Some other specific comments:

Lines 163: "low nitrogen was added: : :" I don't think this is a correct expression of introducing low nitrate concentration. Could the authors also specify how the nitrate concentration was reduced? The same for line 164, "low phosphate was added..".

Response: Agree. We changed 'added' to 'supplied' in lines 163–167.

Line 269: The cell diameter was measured for the whole coccosphere, with coccoliths attached. However, both PIC quota and PIC/POC ratio was changed by different experimental manipulations, especially by alteration of pCO2. This would have also resulted in changes in coccolith thickness. I was wondering if the authors have considered this when calculating the cell-volume normalized particulate organic elemental quotas.

Response: We agree with the suggestions of this referee. We have calculated the cell-volume normalized POC and PIC quotas in figures S6 and S7. POC (or PIC) quota and the cell-volume normalized POC (or PIC) quota showed similar trends in response to different environmental conditions (Figures 4; 5; S6; S7). We added 'and the cell-volume normalized POC quotas' in line 653 in the marked-up manuscript version (below).

Lines 527-531: This sentence is too long, please split to two.

Response: This sentence was reduced to 'Our results from laboratory experiments with multiple drivers experiment instead predicted a different trend with progressive ocean climate changes.' in lines 533–535 in the marked-up manuscript version (below).

Line 556: "low-pH inhibited growth.." Here the authors indicate it was mainly the effects of pH instead of changing pCO2, please add some explanations on this.

Response: We added these contents 'In ocean acidification condition, the negative effect of low pH on growth rate of the same *E. huxeyi* strain PML B92/11 was larger than the positive effect of high $CO_2$ concentration (Bach et al., 2011). Our data further showed that low-pH inhibited growth to lesser extent under the high light than under low light (Fig. 3e; Table 2).' in lines 581–585 in the marked-up manuscript version (below).
.

Bach, L. T., Riebesell, U., and Schulz, K. G.: Distinguishing between the effects of ocean acidification and ocean carbonation in the coccolithophore *Emiliania huxleyi*, Limnol. Oceanogr., 56, 2040–2050, doi: 10.4319/lo.2011.56.6.2040, 2011.

Line 559: Please add a reference after the sentence "photosynthesis under the high light regime could generate more energy-conserving compounds".

Response: Fernández et al. (1996) reported that high light intensity facilitates carbohydrate accumulation and low light intensity reduces cellular carbohydrate content. So we cited this reference in line 587.

Fernández, E., Fritz, J. J., and Balch, W. M.: Chemical composition of the coccolithophorid *Emiliania huxleyi* under light-limited steady state growth. J. Exp. Mar. Biol. Ecol., 207, 149–160.

doi: 10.1016/S0022-0981(96)02657-3, 1996.

Line 561: Please specify the strain of the *E. huxleyi* examined in Jin et al., as well as in line 569.

Response: Jin et al. (2017) examine responses of growth rate, POC and PIC quotas of *E. huxleyi* strains PML B92/11 and CCMP 2090 under different levels of incident solar radiation. '*E. huxleyi*' was replaced by '*E. huxleyi* strains PML B92/11 and CCMP 2090' in line 589, and by '*E. huxleyi* strain PML B92/11' in line 598.

Line 575: Why was PIC quota increased under high light? Please add some explanations.

Response: One explanation could be that high light intensity makes cells to remove $H^+$ faster and then reduce the negative effects of low pH on calcification of *E. huxleyi* (Jin et al., 2017). These contents 'increased light levels can partially counteract the negative effects of OA on calcification' were changed to 'high light intensity could make cells to remove $H^+$ faster and then reduce the negative effects of low pH on calcification of *E. huxleyi* (Jin et al., 2017)' in lines 606–607 in the marked-up manuscript version (below).

Line 617: The sentence "released organic compounds should be negligible: : :" contradicts to the previous expression of "over-synthesis of cellular organic carbon might be released as dissolved organic carbon.." in lines 612-613.

Response: Thanks. We have deleted these contents 'However, released organic compounds should be negligible, since they are usually photorespiration-dependent (Beardall, 1989; Obata et al., 2013)' in lines 666–667.

Fig. 1 Please label the symbols in the graph for a better understanding of the treatments.
Response: Thanks. We have done in Figure 1

Fig. S1 I think it would be better to move this figure to the main manuscript, instead of being in the supplementary materials, in order to make a better understanding of the step-wise experimental design.
Response: We try to move the figure S1 to the main manuscript, whereas we find that figure S1 and figure 1 seem to repeat in terms of the experiment setup. So we would like to keep the figure S1 in the supplemental information. If the referee persists in, we will do it.

Fig. S11 How was the RNA concentration measured? This is not presented in the methods section.
Response: The sentence: 'In this study, RNA content per cell was verified by a SYBR Green method (Berdalet et al., 2005).' is added in line 617–618 in the marked-up manuscript version (below).

Berdalet, E., Roldán, C., Olivar, M. P., and Lysnes, K.: Quantifying RNA and DNA in planktonic organisms with SYBR Green II and nucleases. Part A. Optimisation of the assay, Sci. Mar., 69, 1–16, doi: 10.3989/scimar.2005.69n11, 2005.

**A list of all relevant changes**

**Materials and Methods:**

**Lines 154–155:** Add 'Initial DIN and DIP concentrations were 24 μmol $L^{-1}$ and 1.5 μmol $L^{-1}$, respectively, and initial light intensity was 60 μmol photons $m^{-2}$ $s^{-1}$.'

**Lines 163–164:** Add ', 60 μmol photons $m^{-2}$ $s^{-1}$' and ', 240 μmol photons $m^{-2}$ $s^{-1}$'.

**Lines 164–167:** 'added' is changed to 'supplied'.

**Lines 180–183:** Add 'Such stepwise reduction of nutrients levels would be useful for us to analyze effects of nitrate and phosphate separately, and be expected to have implications for the cells episodically exposed to different levels of nutrients in the sea.'

**Line 239:** 'taken' is changed to 'took'.

**Results**

**Lines 443–444:** Add a sentence: 'However, the opposite results were found under the elevated $CO_2$ treatment alone.'

**Discussion**

**Line 535:** 'change' is changed to 'changes'.

**Lines 535–537:** Delete ', suggesting that some key elements of understanding phytoplankton responses to changing conditions that would enable researchers to connect laboratory studies and field observations are missing.'

**Lines 537–540:** Add these contents: 'We have to admit that results from laboratory experiments can hardly extrapolate to natural conditions. Nevertheless, our data provide mechanistic understanding of the combined effects of ocean climate change drivers, which can be useful in analyzing field observations.'

**Lines 545–560:** Add these contents: 'Different *E. huxleyi* strains displayed optimal responses to a broad range of temperature or $CO_2$ level, and *E. huxleyi* strains isolated from different regions showed local adaptation to temperature or $CO_2$ level (Zhang et al., 2014; 2018). Strain-specific responses of growth, POC and PIC production rates in *E. huxleyi* isolated from different regions to changing seawater carbonate chemistry have also been documented (Langer et al., 2009). It has been suggested that inter-strain genetic variability has greater potential to induce larger phenotypic differences than the phenotypic plasticity of a single strain cultured under a broad range of variable environmental conditions (Blanco-Ameijeiras et al., 2016). On the other hand, the genetic adaptation to culture experimental conditions over time may no longer accurately represent the cells in the sea, as reflected in a diatom (Guan and Gao, 2008). Phytoplankton species that had been maintained under laboratory conditions might have lost original traits and display different responses to environmental changes (Lakeman et al., 2009). The strain used in this study has been kept in the laboratory for about 30 years, and the data obtained in this work can hardly reflect relation to its biogeographic origin.'

**Lines 581–584:** Add these contents: ' In ocean acidification condition, the negative effect of low pH on growth rate of the same *E. huxeyi* strain PML B92/11 was larger than the positive effect of high $CO_2$ concentration (Bach et al., 2011). Our data further showed that'.

**Line 584:** Delete 'Interestingly,'

**Line 587:** Add '(Fernández et al., 1996).', and change 'which' to 'This'.

**Line 589:** Delete 'was' and add 'strains PML B92/11 and CCMP 2090'.

**Line 590:** Add 'were'.

**Line 598:** Add 'strain PML B92/11'.

**Line 600:** Change 'replicate' to 'replete'.

**Lines 605–606:** Delete: 'increased light levels can partially counteract the negative effects of OA on calcification'.

**Lines 606–607:** Add 'high light intensity could make cells to remove $H^+$ faster and then reduce the negative effect of low pH on calcification of *E. huxleyi* (Jin et al., 2017)'

**Lines 616–629:** Add these contents: 'Synthesis of RNA is a large biochemical sink for phosphate in *E. huxleyi* and other primary producers (Dyhrman, 2016). In this study, RNA content per cell was verified by a SYBR Green method (Berdalet et al., 2005). Compared to HNHP conditions, HNLP-grown cells had only 7.8% of total RNA (Fig. S11). This indicates that decreased availability of phosphate strongly decreased RNA synthesis, which would consequently extend the interphase of the cell cycle where calcification occurs (Müller et al., 2008). This could explain why PIC quotas were enhanced by decreased phosphate availability (Fig. 5). Similarly, decreased availability of nitrate decreased protein (or PON) synthesis (Fig. S10), which can also block cells in the interphase of the cell cycle, and increase the time available for calcification in *E. huxleyi* (Vaulot et al., 1987). Consistently with this, lower rates of assimilation or organic matter production in *E. huxleyi* in LNHP than in HNHP treatments are consistent with more energy being reallocated to use for calcification (Nimer and Merrett, 1993; Xu and Gao, 2012).'

**Lines 643–646:** Add these contents: 'While substantial evolutionary responses to multiple drivers may help further, our results imply that decreased phosphate availability along with progressive ocean acidification and warming in surface ocean may reduce the competitive capability of *E. huxleyi* in oligotrophic waters.'

**Line 646:** 'On the other hand' is changed to 'Meanwhile'.

**Line 653:** Add 'and the cell-volume normalized POC quotas'.

**Line 654:** Add 's' and 'S6;'.

**Lines 666–667:** Delete these contents: 'However, released organic compounds should be negligible, since they are usually photorespiration-dependent (Beardall, 1989; Obata et al., 2013).'

**Lines 668–679:** Delete these contents: 'Synthesis of RNA is a large biochemical sink for phosphate in *E. huxleyi* and other primary producers (Dyhrman, 2016). Compared to HNHP conditions, HNLP-grown cells had only 7.8% of total RNA (Fig. S11). This indicates that decreased availability of phosphate strongly decreased RNA synthesis, which would consequently extend the interphase of the cell cycle where calcification occurs (Müller et al., 2008). This could explain why PIC quotas were enhanced by decreased phosphate availability (Fig. 5). Similarly, decreased availability of nitrate decreased protein (or PON) synthesis (Fig. S10), which can also block cells in the interphase of the cell cycle, and increase the time available for calcification in *E. huxleyi* (Vaulot et al., 1987). Consistently with this, lower rates of assimilation or organic matter production in *E. huxleyi* in LNHP than in HNHP treatments are consistent with more energy being reallocated to use for calcification (Nimer and Merrett, 1993; Xu and Gao, 2012).'

**Lines 688–692:** Delete these contents: 'While substantial evolutionary responses to multiple drivers may help further, our results imply that decreased phosphate availability along with progressive ocean acidification and warming in surface ocean may reduce the competitive capability of *E. huxleyi* in oligotrophic waters.'

For step 1, $NO_3^-$ and $PO_4^{3-}$ were modified to 24 μmol L$^{-1}$ and 1.5 μmol L$^{-1}$, respectively, which is the HNHP treatment in the synthetic seawater (Sunda et al., 2005) (Fig. S1). The seawater was dispensed into 4 glass bottles, and 2 bottles of seawater were placed at 16 °C (LT) in an incubator (HP400G-XZ, Ruihua, Wuhan), and aerated for 24 h with filtered (PVDF 0.22 μm pore size, Haining) air containing 400 μatm (LC) or 1000 μatm $p$CO$_2$ (HC). Another 2 bottles of seawater were maintained at 20 °C (HT) in the other chamber and also aerated with LC or HC air as described above. The dry air/CO$_2$ mixture was humidified with deionized water prior to the aeration to minimize evaporation. The LCLT, HCLT, LCHT and HCHT seawaters (Figs. 1a and S1) were then filtered (0.22 μm pore size, Polycap 75 AS, Whatman) and carefully pumped into autoclaved 250 mL polycarbonate bottles (Nalgene, 4 replicate flasks for each of LCLT, HCLT, LCHT and HCHT, a total of 16 flasks at the beginning of the experiment) with no headspace to minimize gas exchange. The flasks were inoculated at a cell density of about 150 cells mL$^{-1}$. The volume of the inoculum was calculated (see below) and the same volume of seawater was taken out from the bottles before inoculation. The samples were initially cultured at 60 μmol photons m$^{-2}$ s$^{-1}$ (LL) of photosynthetically active radiation (PAR)

(measured using a PAR Detector, PMA 2132 from Solar Light Company) under

LCLT, HCLT, LCHT and HCHT conditions for 8 generations (6 days) (d), and then the samples were diluted to their initial concentrations and grown for another 8

generations (6 d) (Fig. 1a). Samples in culture bottles were mixed twice a day at 9:00

a.m. and 5:00 p.m. At the end of the incubation, sub-samples were taken for measurements of cell concentration, POC and TPC quotas, TA, pH and nutrient concentrations.

In step 2, samples grown under the previous conditions were transferred at the end of the cultures from 60 (LL) to 240 $\mu$mol photons m$^{-2}$ s$^{-1}$ (HL) of PAR with initial cell concentrations of 150 cells mL$^{-1}$, and acclimated to the HL for 8 generations (5 d in

16 $^{\circ}$C environment, 4 d in 20 $^{\circ}$C environment) (Fig. 1b). The cultures were then diluted to achieve initial cell concentration and incubated at the HL for another 8

generations (the fifth day in 16 $^{\circ}$C environment and the fourth day in 20 $^{\circ}$C

environment) before sub-samples were taken for measurements.

In step 3, step 4 and step 5, $NO_3^-$ and $PO_4^{3-}$ concentrations were set to be 8 $\mu$mol L$^-$

$^1$ and 1.5 $\mu$mol L$^{-1}$ for the LNHP treatment, and 24 $\mu$mol L$^{-1}$ and 0.5 $\mu$mol L$^{-1}$ for the

[revised manuscript text omitted]

Figure 1

[Figure]

[Figure]

Figure 2

[Figure]

[Figure]

[Figure]

Figure 3

[Figure]

[Figure]

Figure 4

[Figure]

[Figure]

Figure 5

[Figure]

[Figure]

Figure 6